

# Using unmanned aerial vehicle and volunteered geographic information to sophisticate urban flood modelling

Yuan-Fong Su [1], Yan-Ting Lin [2], Jiun-Huei Jang [3], and Jen-Yu Han [4]

[1] Slopeland and Hydrological Hazards Division, National Science and Technology Center for Disaster Reduction, New Taipei City, 231, Taiwan

[2] Department of Civil Engineering, National Taiwan University, Taipei, 106, Taiwan

[3] Department of Hydraulic and Ocean Engineering, National Cheng Kung University, Tainan, 701, Taiwan

[4] Department of Civil Engineering, National Taiwan University, Taipei, 106, Taiwan

*Correspondence to*: Jiun-Huei Jang (jamesjang@mail.ncku.edu.tw)

**Abstract.**

Sophisticated flood simulation in urban areas is a challenging task due to the difficulties in data acquisition and model verification. This study incorporates three rapid-growing technologies, i.e. volunteered geographic information (VGI), unmanned aerial vehicle (UAV), and computational flood

simulation (CFS) to reconstruct the flash flood event occurred in 14 June 2015, GongGuan, Taipei. The high-resolution digital elevation model (DEM) generated by a UAV and the real-time VGI photos acquired from social network are served to establish and validate the CFS model, respectively. The DEM data are resampled based on two grid sizes to evaluate the influence of terrain resolution on flood simulations. The results show that flood scenario can be more accurately modelled as DEM resolution

increases with better agreement between simulation and observation in terms of flood occurrence time and water depth. The incorporation of UAV and VGI lower the barrier of sophisticated CFS and shows great potential in flood impact and loss assessment in urban areas.

**Keywords**: urban flooding; unmanned aerial vehicle; volunteered geographic information; computational flood simulation; social media

## 25    1    Introduction

Flash flooding resulted from extreme heavy rainfall has been recognized as one of the most common and destructive threats in recent years (Panthou et al., 2014; Chan et al., 2016; Bao et al., 2017; Busuioc





et al., 2017; Yang et al., 2017; Fu et al., 2019). Summarized in Table 1 are some of the flash flood events in 2019 which have caused significant economic damages, life losses, and transportation interruption to

many major cities around the world. In the last two decades, computational flood simulation (CFS) has been widely used to generate detailed flood scenarios in space and time by simulating water transportation on surface and in sewer systems (Hunter et al., 2007; Kuiry et al., 2010; Seyoum et al., 2012; Jahanbazi and Egger, 2014; Chang et al., 2015; Jang et al., 2018; Jang et al., 2019). However, these CFS models require detailed DEM (digital elevation model) and real-time flood records for model construction and

validation, which are often inadequate in timeliness and accuracy for flash flood events occurring rapidly in localized areas (Suarez et al., 2005; Yin et al., 2015; Pregnolato et al., 2017).

Many researches have highlighted the influence of DEM resolution on hydrological modelling (Vaze et al., 2009; Leitao et al., 2009; Li and Wong, 2010; Saksena and Marwade, 2015). For urban flood modelling, Yang et al. (2014) recommended that the resolution of DEM should be higher than 5 m to

properly represent topographical indices. Flood simulation under coarser DEM resolutions tend to overestimate the flood area and underestimate the flood depth in low-lying urban areas (Kim et al., 2020). Recently, some authors mentioned that high resolution DEM has a greater influence on flood damage estimation than on flood hazard estimation (Komolafe et al., 2018). Low spatial resolutions may generate large errors in flood loss estimation due to improper representation of buildings (Afifi et al., 2009). Thus,

the acquisition of high resolution DEM has become a crucial task for sophisticated flood impact analysis in urban areas.

Traditionally, the DEM data are derived by airborne Lidar which are too costly to be updated frequently (Sankey et al., 2018), and flood records are usually obtained by post-disaster field investigations that contain only rough coordinates and water depths without detailed time series. Recently,

two raising techniques namely unmanned aerial vehicle (UAV) and volunteered geographic information (VGI) have been adopted for DEM generation and flood detection, respectively (Le Coz, et al., 2016; Michelsen, et al., 2016; Starkey et al., 2017; Tauro et al., 2018). Studies have shown that digital elevation models (DEM) derived by UAV have similar performances in urban overland flow modelling compared



with that derived from Lidar (Leitão et al., 2016). The VGI considers every citizen as a sensor to acquire

spatial data on a wide range of phenomena via crowdsourcing the keywords on social media such as

Facebook, Twitter, Instagram, etc. (Goodchild and Glennonm, 2010). Cervone et al. (2015) used Twitter

for remote sensing data collection and damage assessment of transportation infrastructure in the case

study of 2013 Boulder flood. Huang et al. (2018) proposed a convolutional neural network (CNN)

architecture to classify the flood pictures and a sensitivity test to extract flood-sensitive keywords that

were further used to refine the CNN results.

The applications of UAV and VGI open a new page for the sophistication of CFS models. Compared

with traditional methods, the UAV and VGI are more economical and applicable to retrieve detailed

terrain and flood information in real-time. The DEM generated by UAV can be served as the boundary

conditions to increase the spatial resolution of CFS and the time-series of water levels retrieved by VGI

can be used to validate the temporal variation of CFS results. With the help of UAV and VGI, this paper

introduces the methodologies and demonstrates the advantages of conducting high-resolution CFS for

flood analysis in urban areas, which are crucial for impact and loss assessment.

## 2   Materials and methods

The flash flood event occurred on 14 June 2015 in GongGuan, Taipei, Taiwan, is selected for case

study. The rainfall event occurred between 13:00-18:00 on 14 June 2015 with hourly rainfall peak of

131.5 mm/h during 14:30-15:30 as shown in Fig. 1. This rainfall intensity has exceeded the designed

drainage capacity of the sewer system and resulted in severe flash flooding in the cross-section of Keelung

Road and Changxing Street nearby the National Taiwan University. The study area and the location of the

rain gauge are shown in the Fig. 2. The DEM derived by UAV and the flood photos collected from VGI

are served to establish and validate the CFS, respectively. The conceptual flowchart of this study is shown

in Fig. 3. First, the UAV is deployed in a clear weather after the flood event to collect a great number of

images for generating DSM/DEM of the study area. Second, the rainfall and DEM under two different

resolutions are introduced into a CFS model to reconstruct the time series of flood depth and extent for

the selected flood event. Finally, the simulated results are compared with the VGI photos to see the



influence of DEM resolution on CFS.

## 2.1 DEM generated by UAV

The procedure of developing the urban 3D terrain is shown on the left side of Fig. 1. The methods of
generating DEM from a set of aerial images or videos are quite mature (Pollefeys et al., 2008; Zhou et al.,
2004), which are based on the fundamental principle of collinearity condition (Fig. 4) expressed by the

following equations:

$$x_p = -f \left[ \frac{m_{11}(X_P - X_L) + m_{12}(Y_P - Y_L) + m_{13}(Z_P - Z_L)}{m_{31}(X_P - X_L) + m_{32}(Y_P - Y_L) + m_{33}(Z_P - Z_L)} \right] \tag{1}$$

$$y_p = -f \left[ \frac{m_{21}(X_P - X_L) + m_{22}(Y_P - Y_L) + m_{23}(Z_P - Z_L)}{m_{31}(X_P - X_L) + m_{32}(Y_P - Y_L) + m_{33}(Z_P - Z_L)} \right] \tag{2}$$

where $x_p$ and $y_p$ are the image coordinate of any point $p$; $X_P, Y_P, Z_P$ represent the ground coordinate
of point $p$; $X_L, Y_L, Z_L$ represent ground coordinate of the projection (optical) centre; $f$ is the focal length;
$m_{11} \ldots m_{33}$ are the coefficients of a $3 \times 3$ rotation matrix defined by the angles $\omega$, $\phi$, and $\kappa$ that
transforms the ground coordinate system to the image coordinate system (Lillesand and Kiefer, 1999).

The six parameters $X_L, Y_L, Z_L$ and $\omega$, $\phi$, and $\kappa$ are used for exterior orientation of an image, which can
be determined through the process of space resection as illustrated in Fig. 4. In the figure, the $X$, $Y$, and
$Z$ coordinate of any point in the matched stereopair of tilted images can be determined. Using a set of
images taken by UAV, the ground coordinate of any point in the overlap of tilted images can be determined
by the image matching method of structure-from-motion (Remondino and Fraser, 2006; Westoby et al.,

2012). Finally, the coordinate accuracies at check points are examined and the urban 3D terrain is obtained
by resampling the coordinates to a regular gird system by the nearest neighbouring method.

The UAV used in this study is DJI Phantom 2 Vision+ (Da-Jiang Innovations) which weights 1.2 kg
and has a camera with 4384×2466 pixels. The focal length of the camera is 3.3 mm and the field-of-view
is 110 degrees. The UAV campaign was conducted in the early morning 22 July 2015 in cloudless

condition during 06:00 to 06:40 to reduce the disturbances from weather and traffic. In total, 589





positioned images were acquired with an overlap ratio of 75%–85% and a mean spatial resolution of 2.84

cm. After space intersection, the average ground sampling distance of point cloud is 0.03m. The UAV

images were processed to generate orthomosaic image and digital surface model (DSM) with

Pix4Dmapper Pro Version 1.4.46 (Pix4D). To derive absolute coordinates, three ground control points

(GCPs) were distributed in the study area (Fig. 5). The coordinates of the three GCPs were acquired using

the static positioning of Global Navigation Satellite System (GNSS) with positional accuracy in

centimeter level. The absolute positions of the images, captured through the GNSS receiver in the UAV,

were recorded to establish the coordinate system constrained on the three GCPs.

The lens distortion of the camera was calibrated by the flexible and powerful self-calibrating bundle

adjustment (Remondino and Fraser, 2006). The calibration relies on the continuous overlapped-images so

as to do the aerotriangulation adjustment. After the interior orientation and the coordinates of the object

points are calculated, the correction $\Delta x$ and $\Delta y$ can be revised as below:

$$\begin{cases} \Delta x = -\Delta x_0 - \dfrac{x_i}{f}\Delta f + K_1 x_p r^2 + K_2 x_p r^4 + K_3 x_p r^6 + P_1\left(r^2 + 2x_p^2\right) + 2P_2 x_p y_p \\ \Delta y = -\Delta y_0 - \dfrac{y_i}{f}\Delta f + K_1 y_p r^2 + K_2 y_p r^4 + K_3 y_p r^6 + P_2\left(r^2 + 2y_p^2\right) + 2P_1 x_p y_p \end{cases} \tag{3}$$

where $\Delta f$ is the principal distance error, $\Delta x_0$ and $\Delta y_0$ are the displacements of the principal point, $K_1$,

$K_2$, and $K_3$ are the parameters of the radial distortion, $P_1$ and $P_2$ are the parameters of the decentering

distortion, and $r$ is the distance between the image point and the principal point. The calibrated

parameters are listed in Table 2.

For CFS application, the DSM was converted to DEM by removing the vegetation and the viaduct.

The vegetation such as shrubs and grasses is detected by the normalized difference vegetation index

(NDVI) in the range of 0.2–0.3 (Candiago et al., 2015). Since the UAV images only observed red, green

and blue bands, the near infrared band was built on a specific linear combination of the three bands with

a lower-pass filter (Rabatel et al., 2011). To remove the viaduct, the elevation higher than the Keelung

Road (9 m) was selected as a threshold value to identify the viaduct positions. Based on the NDVI and



elevation thresholds, the vegetation and viaducts were filtered out in DEM so that flood water can transport smoothly on ground surface. However, unlike traditional DEM, the elevations of buildings were

not removed to prevent the transverse of runoff water.

## 2.2 VGI from social media

Ethical and legal concerns are big issues for collecting and using VGI (Foody et al., 2017). Fortunately, the Copyright Act and Relative Laws in Taiwan allows researchers to quote, within a reasonable scope, publicly released works for reports, comment, teaching, research or other legitimate purposes (Copyright

Act and Relative Laws, 2016). Based upon the Act, the VGI data used in this study are collected from the most famous Bulletin Board System (BBS) in Taiwan named PTT. There are 8 photos collected from PTT posted during 15:20~16:30 on 14 June 2015. From these photos, we visually identified 8 locations in the study area as shown in Fig. 6. The timestamp and the virtual water depths in these photos are served to validate the CFS model. Although the timestamp when photos were posted on internet may not always be

the acquisition time and the flood depth estimated from photos may be subject to experts' experience, these uncertainties can be reduced by the functions of "live stream" and "image recognition" on social media

## 2.3 CFS model

The CFS model used in this study was developed by Jang et al. (2018), in which a 2D Overland Flow

Model (OFM) is coupled with a 1D Sewer Flow Model (SFM) for sophisticated flood simulation in cities with drainage systems. The OFM and SFM are established based on shallow water equations and finite difference numerical methods. The alternate direction explicit scheme and implicit backward Euler algorithm are used to solve the OFM and SFM, respectively. The SFM adopted Preissmann slot method (Cunge and Wegner, 1964) to calculate the full and partially-full flow conditions at the same time. For

the OFM simulation, the elevation at each grid center is extracted from the DEM data. When rain drops, the OFM is firstly initiated for surface water routing and the SFM is then initiated by the water that flows into the sewer pipes via street inlets. When the sewer pipes get full, the sewer water surcharges back onto

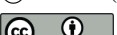



ground surface via manholes. In the simulation process, the water exchanged between the two models are

determined by weir and orifice functions via one-to-one relationship. The details of the CFS model can

be referred to Jang et al. (2018; 2019).

## 3    Results and discussions

### 3.1 DEM

The DEM accuracy are examined at the three GCPs. The errors in $X$, $Y$ and $Z$ directions range

from -0.006 to 0.025 m, -0.038 to 0.061 m, and -0.016 to 0.002 m, respectively. The root mean square

errors of the absolute coordinate at the three GCPs are less than 0.046m (Table 3). The orthoimage and

the DEM resampled under spatial resolutions of 0.5 m and 5 m are shown in Fig. 7. In the figure, the

elevation distributes from 5.5 to 55.8 m and the buildings are displayed in the warm colors.

### 3.2 Flood extent

To discover the influence of DEM resolution on flood simulation, the gird meshes of the CFS model

are established under a fine grid size ($0.5 \ \mathrm{m} \times 0.5 \ \mathrm{m}$) and a coarse grid size ($5 \ \mathrm{m} \times 5 \ \mathrm{m}$), respectively,

in which the elevation at each grid center is extracted from the DEM with accordant resolution. The flood

extents simulated under the two DEM resolutions at different time are displayed in Fig. 8, in which the

VGI points out of the 8 locations are marked if the simulated flood depths exceed 0.05 m. The value of

0.05 m is selected as the threshold because above which road traffic will be affected. In the case with 0.5

m DEM resolution, flooding starts around 14:00 at points #4, #7, and #8, peaks around 15:00 at all points,

and retreats at 17:00 back to original points. In comparison, flooding simulated under 5 m DEM resolution

occurs earlier and retreats much later with some water trapped between buildings. Observing the flood

maps between 15:00-18:00, the water inundated on the rooftops of buildings can be properly simulated

in the case with fine DEM resolution but not in the case with coarse resolution.

### 3.3 Flood volume

The time series of flood volume simulated under the two DEM resolutions are displayed in Fig. 9.



Compared with the case with 0.5 m DEM resolution, the flood volume simulated under 5 m DEM resolution arises faster but descends slower with a higher and earlier appearance of flood peak. This implies that, when DEM resolution decreases, the topography becomes rugged, the friction increases, and

the flood water travels slower. Table 4 compares the simulation and observation results in terms of flood occurrence time and water depth. The timestamps and estimated water depths (WD) are obtained from the VGI photos in Fig. 6, and the flood durations at the eight VGI points when the water depth exceeds 0.05 m are determined based on the CFS results. It is seen that the timestamps of VGI photos all lie within the simulated flood duration at the points with observed WD larger than 0.05 m (points #1, #2, #4, #7,

and #8). At the rest points, the simulated and observed WDs are both smaller than 0.5 m. This good agreement between observation and simulation reveals that the flood model is accurate in rebuilding the process of flood transport under both DEM resolutions.

Compared with the flood durations simulated under 0.5 m DEM resolution, those under 5 m DEM resolution increase by 30 to 200 minutes at points #1, #2 and #3. This indicates a chance that the flood

durations are overestimated under 5 m DEM resolution because no further VGI photos were posted after 15:40 at point #2. The flood impact on traffic could be overestimated if low-resolution DEM data are used for flood simulation. The spatial resolution of DEM in urban areas should be at least finer than road width so that road profiles can be clearly displayed; otherwise, runoff transportation around buildings and on roads cannot be correctly simulated. Sub-meter resolution DEM data generated by UAV are adequate for

assessing the impact of localized flooding on transportation in cities. Some open accessible topographic datasets, such as the 30 m resolution DEM by STRM from NASA (https://www2.jpl.nasa.gov/srtm/) and the open DEM in Taiwan with 20 m resolution (https://data.gov.tw/dataset/35430) are too coarse to serve the purpose.

### 3.4 Simulation efficiency

Flood simulation under high grid resolutions is usually more time-consuming than that that under coarse grid resolutions due to the increase in grid numbers. The choice of grid resolution for flood simulation is a tradeoff between accuracy and efficiency. For the study area in this research, there are

573,000 and 5,730 grids for the mesh with grid size 0.5 m and the mesh with grid size 5 m, which

consumes 1,127 mins and 16 mins of computational time, respectively (with Intel Core i7-7700K CPU

@ 4.2GHz and 64 GB RAM). For disaster emergency response in regional scale, flood simulation under

coarse grid resolution is enough to gain a fast and overall understanding of flood pattern. However, for

evaluating flood impact on critical infrastructures such as metro stations, power facilities, schools,

government agencies, hospitals, etc., high-resolution flood simulation is required.

## 4    Conclusions

Flash flood analysis by CFS in urban area is a challenging task since it requires high-resolution terrain

and real-time flood information for model construction and validation. Aided by the rapid growing

technologies of remote sensing and crowdsourcing, it is possible to update DEM data and record the flood

depth in real time by UAV and VGI. In this study, we adopt the UAV and VGI to sophisticate CFS

modeling in the reconstruction of a flash flood event occurred on 14 June, 2015, Taipei City. The CFS

model is routed under two DEM resolutions with grid size 0.5 m and 5 m, separately. In the case with

finer DEM, the simulation results are in better agreement with the observation in terms of flood extent,

depth, and occurrence. Using DEM with coarse resolution for CFS overestimates the flood duration on

roads which may provide bias information to decision makers for impact assessment on traffic and

economic losses. Compared with traditional methods, the UAV and VGI are more economical and

applicable in acquiring necessary data for high-resolution CFS.

*Data availability*. DEM and VGI data used in this study are available by contacting the authors.

*Author contributions*.

JHJ and YFS conceived the idea; YTL and JHJ conducted the experiments and analyses; YFS, JHJ, and

YTL wrote the article; JYH provided comments.

*Competing interests*. The authors declare that they have no conflict of interest.



*Acknowledgements*. This work was supported by the Ministry of Science and Technology [MOST 108-
2625-M-006-008].

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

Copyright Act 2016, Intellectual Property Office, Ministry of Economic Affairs, ROC., rom:
https://www.tipo.gov.tw/public/Attachment/71417531682.pdf, last access: 26 September, 2019.

Cunge, J.A. and Wegner, M.: Intégration numérique des équations d'écoulement de barré de Saint-Venant



par un schéma implicite de différences finies, La Houille Blanche, 1, 33-39, https://doi.org/10.1051/lhb/1964002, 1964.

Floodlist, http://floodlist.com/, last access: 16 Jan, 2020.

Foody, G., See, L., Fritz, S., Mooney, P., Olteanu-Raimond, A-M., Fonte, C.C., and Antoniou, V.: Mapping and the Citizen Sensor. London: Ubiquity Press, DOI: https://doi.org/10.5334/bbf. License: CC-BY 4.0., 2017.

Fu, J.C., Huang, H.Y., Jang, J.H., and Huang, P.H.: River stage forecasting using multiple additive regression trees, Water Res. Manag. 33, 4491–4507, https://doi.org/10.1007/s11269-019-02357-x, 2019.

Goodchild, M. F. and Glennonm, J. A.: Crowdsourcing geographic information for disaster response: A research frontier, Int. J. Digit. Earth, 3, 231–241, https://doi.org/10.1080/17538941003759255, 2010.

Huang, X., Wang, C., Li, Z., and Ning, H.: A visual-textual fused approach to automated tagging of flood-related tweets during a flood event, Int. J. Digit. Earth, 1248–1264, https://doi.org/10.1080/17538947.2018.1523956, 2018.

Hunter, N.M., Bates, P.D., Horritt, M.S., and Wilson, M.D.: Simple spatially-distributed models for predicting flood inundation: A review, Geomorphology 90(3–4), 208–225, https://doi.org/10.1016/j.geomorph.2006.10.021, 2007.

Jahanbazi, M. and Egger, U.: Application and comparison of two different dual drainage models to assess urban flooding. Urban Water J., 11, 584–595, https://doi.org/10.1080/1573062X.2013.871041, 2014.

Jang, J.H., Chang, T.H., and Chen., W.B.: Effect of inlet modelling on surface drainage in coupled urban flood simulation. J. Hydrol., 562: 168–180., https://doi.org/10.1016/j.jhydrol.2018.05.010, 2018.

Jang, J.H., Hsieh C.T., and Chang, T.H.: The importance of gully flow modelling to urban flood simulation, Urban Water J., 16(5): 377-388, https://doi.org/10.1080/1573062X.2019.1669198, 2019.

Kim, Y.D., Tak, Y.H., Park, M.H., Kang, B.: Improvement of urban flood damage estimation using a high-resolution digital terrain. J. Flood Risk Manag.; 13 ( Suppl. 1):e12575, https://doi.org/10.1111/jfr3.12575, 2020

Komolafe, A., Herath, S. and Avtar, R.: Sensitivity of flood damage estimation to spatial resolution. J.



Flood Risk Manag., 11: S370-S381, https://doi.org/10.1111/jfr3.12224, 2018

Kuiry, S. N., Sen, D., and Bates, P. D.: A coupled 1D-quasi 2D flood inundation model with unstructured
grids, J. Hydraul. Eng. 136, 493–506, https://doi.org/10.1061/(ASCE)HY.1943-7900.0000211, 2010.

Le Coz, J., Patalano, A., Collins, D., Guillén, N.F., García, C.M., Smart, G.M., Bind, J., Chiaverini, A.,
        Le Boursicaud, R., and Dramais, G.: Crowdsourced data for flood hydrology: Feedback from recent
        citizen science projects in Argentina, France and New Zealand. J. Hydrol., 541, 766-777,
        https://doi.org/10.1016/j.jhydrol.2016.07.036, 2016.

Leitão, J.P. and Boonya-aroonnet, S. and Prodanović, D. and Maksimović, Č.: The influence of digital
        elevation model resolution on overland flow networks for modelling urban pluvial flooding. Water
        Sci Technol; 60 (12): 3137–3149, https://doi.org/10.2166/wst.2009.754, 2009

Leitão, J.P., Vitry, M.M.d., Scheidegger, A., and Rieckermann, J.: Assessing the quality of digital elevation
        models obtained from mini unmanned aerial vehicles for overland flow modelling in urban areas,
Hydrol. Earth Syst. Sci., 20, 1637–1653, https://doi.org/10.5194/hess-20-1637-2016, 2016.

Li, J., and Wong, D.: Effect of DEM sources on hydrologic applications. Comput. Environ. Urban, 34.
        251-261, https://doi.org/10.1016/j.compenvurbsys.2009.11.002, 2010.

Lillesand, T.M. and Kiefer, R.W. (4th Ed.): Remote Sensing and Image Interpretation, New York, Wiley.,
        1999.

Michelsen, N., Dirks, H., Schulz, S., Kempe, S., Al-Saud, M., and Schüth, C.: Youtube as a crowd-
        generated    water    level    archive,    Sci.    Total    Environ.,    568,    189-195,
        https://doi.org/10.1016/j.scitotenv.2016.05.211, 2016.

Panthou, G., Vischel, T., and Lebel, T.: Recent trends in the regime of extreme rainfall in the Central Sahel.
        Int. J. Climatol., 34, 3998-4006., https://doi.org/10.1002/joc.3984, 2014.

Pollefeys, M., Nister, D., Frahm, J.-M., Akbarzadeh, A., Mordohai, P., Clipp, B., Engels, C.: Detailed
        real-time    urban    3D    reconstruction    from    Video.    Int.    J.    Comput.    Vision,    78,    143-167,
        https://doi.org/10.1007/s11263-007-0086-4, 2008.

Pregnolato, M., Ford, A., Wilkinson, S.M., and Dawson, R.J.: The impact of flooding on road transport:
        A    depth-disruption    function.    Transpor.    Res.    D-TR    E.,    55,    67-81,



305        https://doi.org/10.1016/j.trd.2017.06.020, 2017.

Rabatel, G., Gorretta, N., and Labbé S.: Getting NDVI Spectral Bands from a Single Standard RGB Digital Camera: A Methodological Approach. In: Lozano J.A., Gámez J.A., Moreno J.A. (Eds.): Advances in Artificial Intelligence. CAEPIA 2011. Lecture Notes in Computer Science (pp. 333-342), vol. 7023. Springer, Berlin, Heidelberg: La Laguna, Spain, 2011.

Remondino, F., and Fraser, C.: Digital camera calibration methods: considerations and comparisons. International Archives of Photogrammetry, Remote Sensing and Spatial Information Sciences, 36, 266-272, http://citeseerx.ist.psu.edu/viewdoc/download?doi=10.1.1.67.8805&rep=rep1&type=pdf, 2006.

Saksena, S., and Merwade, V.: Incorporating the effect of DEM resolution and accuracy for improved

flood inundation mapping. J. Hydrol., 530, 180–194, https://doi.org/10.1016/j.jhydrol.2015.09.069, 2015.

Sankey, T.T., McVay, J., Swetnam, T.L., McClaran M.P., Heilman, P., Nichols, M.: UAV hyperspectral and lidar data and their fusion for arid and semi-arid land vegetation monitoring, Remote Sen. Eco. Conserv., 4:20–33, 2018.

Starkey, E., Parkin, G., Birkinshaw, S., Large, A., Quinn, P., and Gibson, C.: Demonstrating the value of community-based ('citizen science') observations for catchment modelling and characterization, J. Hydrol., 548, 801-817, https://doi.org/10.1016/j.jhydrol.2017.03.019, 2017.

Suarez, P., Anderson, W., Mahal, V., and Lakshmanan, T.R.: Impacts of flooding and climate change on urban transportation: A systemwide performance assessment of the Boston Metro Area. Transpor.

Res. D-TR E., 10, 231-244, https://doi.org/10.1016/j.trd.2005.04.007, 2005.

Seyoum, S.D., Vojinovic, Z., Price, R.K., and Weesakul, S.: Coupled 1D and Noninertia 2D flood inundation model for simulation of urban flooding. J. Hydraul. Eng. 138, 23–34, https://doi.org/10.1061/(ASCE)HY.1943-7900.0000485 , 2012

Tauro, F., Selker, J., van de Giesen, N., Abrate, T., Uijlenhoet, R., Porfiri, M., Manfreda, S., Caylor, K.,

Moramarco, T., and Benveniste, J.: Measurements and observations in the XXI century (MOXXI): innovation and multi-disciplinarity to sense the hydrological cycle. Hydrolog. Sci. J., 63, 169–196,



https://doi.org/10.1080/02626667.2017.1420191, 2018.

Vaze, J., Teng, J., and Spencer, G.: Impact of DEM accuracy and resolution on topo-graphic indices. Environ. Model. Softw. 25, 1086–1098, https://doi.org/10.1080/014311600209931, 2010.

Westoby, M.J., Brasington, J., Glasser, N.F., Hambrey, M.J., and Reynolds, J.M.: Structure-from-Motion photogrammetry: A low-cost, effective tool for geoscience applications. Geomorphology, 179, 300–314, https://doi.org/10.1016/j.geomorph.2012.08.021, 2012.

Yang, P., Ames, D. P., Fonseca, A., Anderson, D., Shrestha, R., Glenn, N. F., and Cao, Y.: What is the effect of LiDAR derived DEM resolution on large-scale watershed model results? Environ. Model.

Softw., 58, 48–57, https://doi.org/10.1016/j.envsoft.2014.04.005, 2014.

Yang, P., Ren, G., and Yan, P.: Evidence for a strong association of short-duration intense rainfall with urbanization in the Beijing urban area. J. Clim, 5851-5870, https://doi.org/10.1175/JCLI-D-16-0671.1, 2017.

Yin, J., Yu, D., Yin, Z., Liu, M., and He, Q.: Evaluating the impact of risk of pluvial flash flood in intra-

urban road network: A case study in the city center of Shanghai, China. J. Hydro., 537, 138-145, https://doi.org/10.1016/j.jhydrol.2016. 03.037, 2016.

Zhou, G., Song, C., Simmers, J., and Cheng, P.: Urban 3D GIS from LiDAR and digital aerial images. Computer Geosciences, 30, 345-353, https://doi.org/10.1016/j.cageo.2003.08.012, 2004.





Table 1: Major flash flood events around the world in 2019 (Floodlist, 2019).

| Location and Date | Description |
|---|---|
| Tafalla, Spain<br>08, July | A rainfall record of 100.2 mm in 24 hours with highest hourly rainfall of 63.8 mm is observed in Tafalla, Spain. Rail transportation and major roads such as N-121 and AP-15 are interrupted. |
| Washington, USA<br>08, July | About 127mm rainfall downpoured in 24 hours in Frederick, nearby Washington D.C. and the Reagan National Airport in Washington D.C. reported 83.82mm of rain in 1 hour on 08 July. Around 20 roads in the D. C. area were closed and several Amtrak trains were stopped due to flooded tracks near Alexandria, Virginia. |
| Texas, USA<br>24, June | More than 317.5 mm of rain pounded Harlingen, Cameron, Texas, USA within 4 hours. Several streets were flooded and blocked. |
| Recife, Brazil<br>13, June | The city recorded 117mm of rain within 6 hours and it is roughly one-third of the monthly average of June. |
| London, England<br>11, June | The area nearby London received 1~2 months worth of rain in 24 hours. Heavy rain has caused widespread disruption in England, including flooded roads, railways, and a hospital. |
| Hesse, Germany<br>21, May | In Hesse and North Rhine-Westphalia up to 50 mm of rain fell in 6 hours causing flooding of roads and car accident. |
| Hsinchu, Taiwan<br>17, May | A rainfall record of 341mm observed in 6 hours with highest hourly rainfall of 91.5 mm. Some major roads in Hsinchu county were blocked due to flooding. |
| Yonaguni, Japan<br>13, May | The highest rainfall record of 367 mm in 6 hours in 50 years was recorded in Yonaguni, Okinawa. Airport was temporally closed and transportation was interrupted. |
| Texas and Kansas, USA<br>07, May | A record of 177.8mm of rain in 4 hours observed in Texas and 127mm of rain in 2 hours observed in Kansas. Major roadways were impassable. |
| Abijan, Ghana<br>08, April | A record of 53mm of rain in 24 hours observed on Abijan and several areas of the city were flooded, including Abosseyokai, Awudome, Kwame Nkrumah Interchange, Kaneshie, Ashaiman, and Alajo. Scores of vehicles were trapped in the water. |
| Lorestan, Iran<br>31, March | In a 24 hours period from 31 March, 2019, Khorramabad, capital of Lorestan Province recorded 106.9mm of rain and Hamedan in Hamadan Province, recorded 98.6mm. The flooding has caused damage to bridges, roads, infrastructure and homes in Lorestan province. |
| Shiraz, Iran<br>25, March | A major flash flooding hit the city of Shiraz after 18mm of rain fell in 10min. Major roadways were flooded and over 150 vehicles dragged by the powerful flood waters. |
| San Paulo, Brazil<br>10, March | Nearly 110mm of rain, 70% of the month worth of March, fell within 24 hours and the flooding water tossed cars atop buildings and into trees. |
| Daintree, New Zealand<br>28, January | The Daintree River village of Whyanbeel Valley measured a record 472mm in 24 hours which worth a month of rainfall in Jan. Major roadways and Ferry were closed due to flooding. |
| Beira, Mozambique<br>22, January | About 277mm of rain was reported in Beira in the 24 hours, more than 250mm expected in the entire month of January, due to Cyclone Desmond. Major roadways were flooded and scores of cars were submerged. |
| Chaco, Argentina<br>08, January | Resistencia, Chaco recorded 224 mm of rain within 24 hours while 180mm of rain fell in just 80 minutes. Major roadways were flooded. |





Table 2: Calibrated parameters for camera on the UAV.

| Parameters | | Value | Parameters | | Value |
|---|---|---|---|---|---|
| Image size (pix) | | 4384*2466 | Focal length(mm) | | 3.32347 |
| Pixel size (µm) | | 1.3306 | Radial distortion | $K_1$ | -0.382174 |
| Principal point (mm) | $x_0$ | 2.9718 | | $K_2$ | 0.182175 |
| | $y_0$ | 1.71489 | | $K_3$ | -0.046338 |
| CCD size (mm) | Width | 5.83335 | Decentering distortion | $P_1$ | 0.000508 |
| | Height | 3.28126 | | $P_2$ | 0.000017 |

Table 3: Accuracy of the ground control points (m).

| | GCP_1 | GCP_2 | GCP_3 | Mean | Std. | RMSE |
|---|---|---|---|---|---|---|
| Error X | -0.006 | 0.025 | -0.019 | 0.000 | 0.018 | 0.018 |
| Error Y | 0.061 | -0.035 | -0.038 | -0.004 | 0.046 | 0.046 |
| Error Z | 0.002 | -0.016 | 0.000 | -0.005 | 0.008 | 0.009 |

Table 4: Comparison between CFS and VGI results.

| Point (#) | Latitude (degree) | Longitude (degree) | Observation (VGI) | | Flood duration with WD ≥ 0.05 m (CFS) | |
|---|---|---|---|---|---|---|
| | | | Timestamp | WD (m) | DEM 0.5 m | DEM 5 m |
| 1 | 121.544603 | 25.016963 | 15:20 | ≥ 0.05 | 14:40–15:30 | 14:00–15:50 |
| 2 | 121.543563 | 25.016420 | 15:40 | ≥ 0.05 | 15:00–15:50 | 13:50–18:00 |
| 3 | 121.544612 | 25.017167 | 16:10 | < 0.05 | 15:00–15:30 | 14:40–15:40 |
| 4 | 121.544236 | 25.017421 | 16:10 | ≥ 0.05 | 14:00–18:00 | 13:50–18:00 |
| 5 | 121.543351 | 25.017728 | 16:20 | < 0.05 | 14:40–16:10 | 14:40–16:00 |
| 6 | 121.543462 | 25.017716 | 16:20 | < 0.05 | 14:40–15:40 | 14:40–15:40 |
| 7 | 121.543674 | 25.017670 | 16:30 | ≥ 0.05 | 14:00–17:50 | 14:00–18:00 |
| 8 | 121.543476 | 25.016745 | 16:30 | ≥ 0.05 | 14:00–18:00 | 13:50–18:00 |



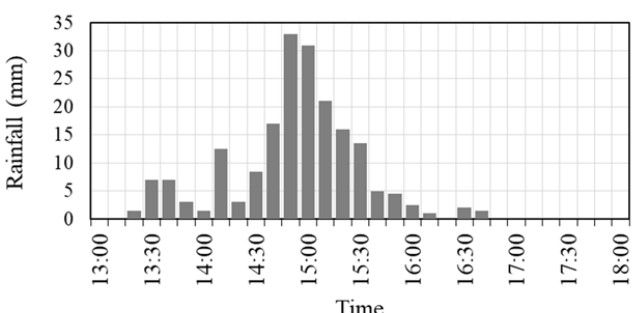

Figure 1: The rainfall hyetograph on 14 June 2015 at GongGuan rain-gauge
station (C1A760).

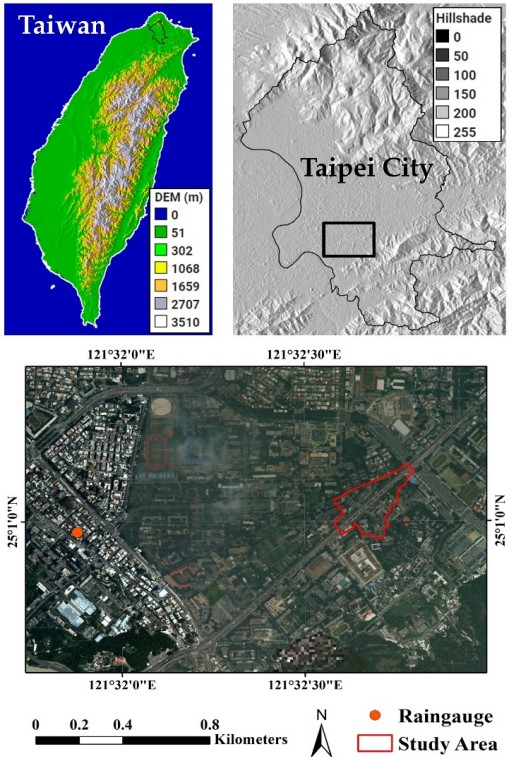

Figure 2: Study area (red polygon) at GongGuan, Taipei, Taiwan (the Google
Earth images sourced from © Google, Landsat/Copernicus and the shading image
was derived from SRTM with 30 meter resolution).





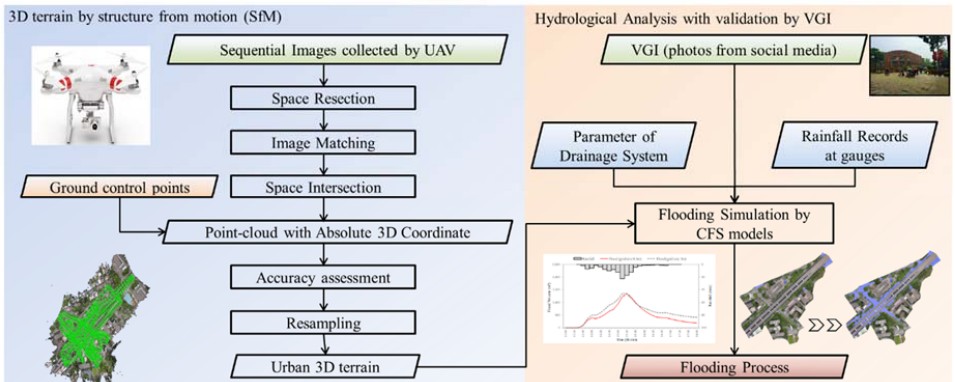

Figure 3: Conceptual flowchart of this study (the VGI photo was adopted from PTT, Taiwan).

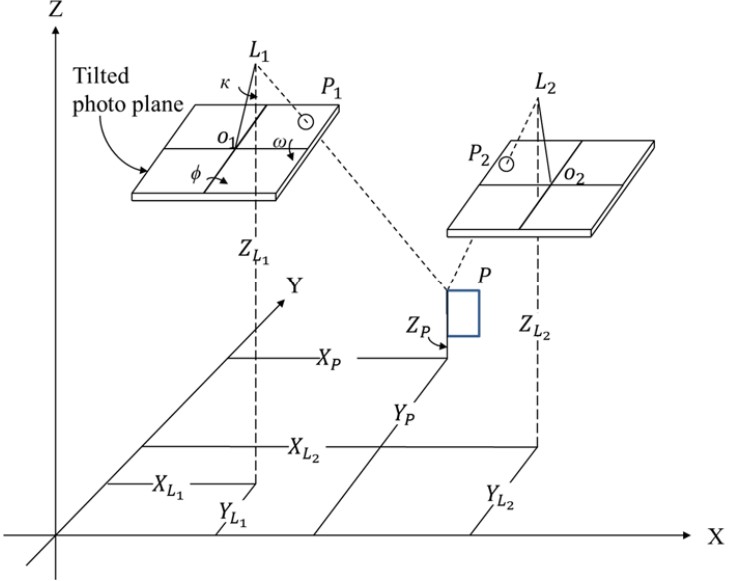

Figure 4: Illustration of collinearity condition and space intersection (adapted from Lillesand and Kiefer, 1999).



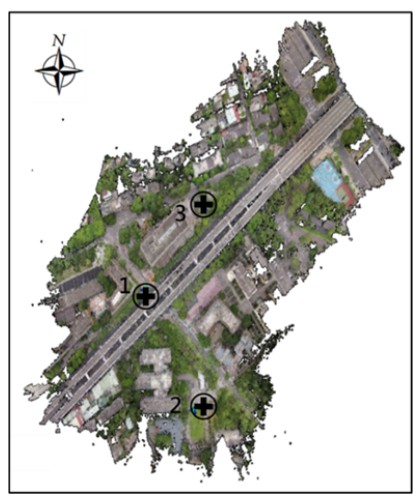

Figure 5: Images taken by UAV and the distribution of the ground control points.

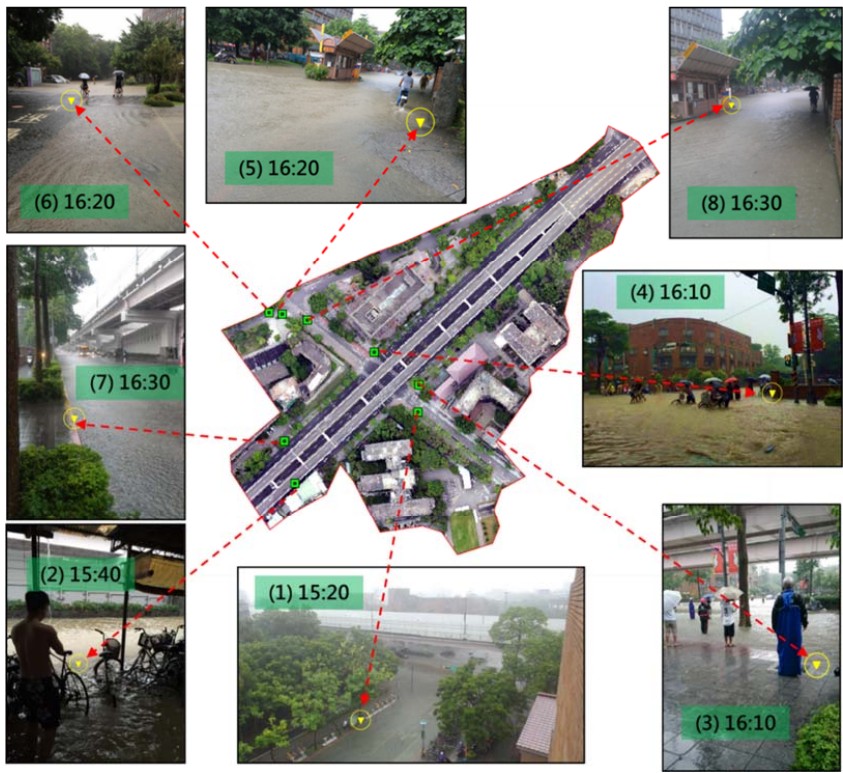

Figure 6: VGI photos from social media and their acquisition time (adopted from

PTT, Taiwan).



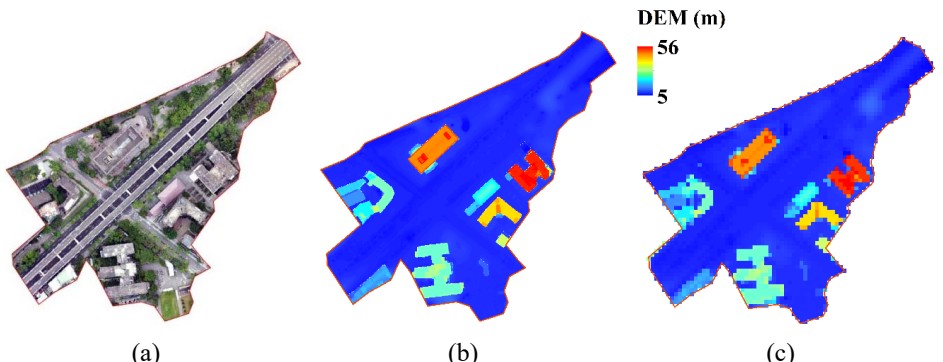

Figure 7: (a) Orthoimage; (b) the DEM with spatial resolution of 0.5 m; (c) the DEM with spatial resolution of 5 m

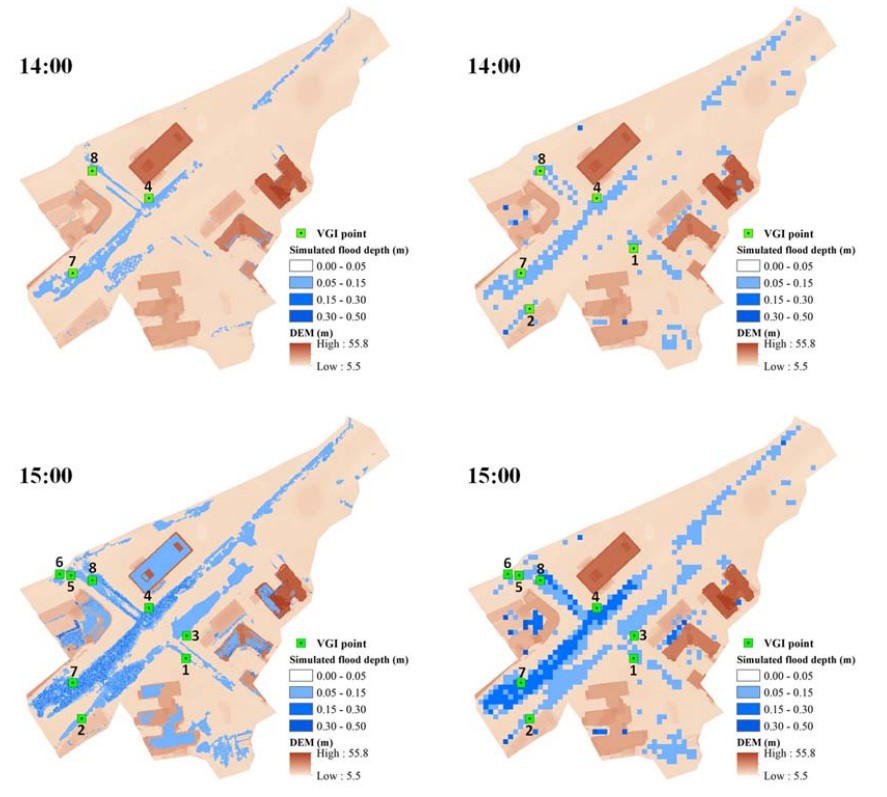

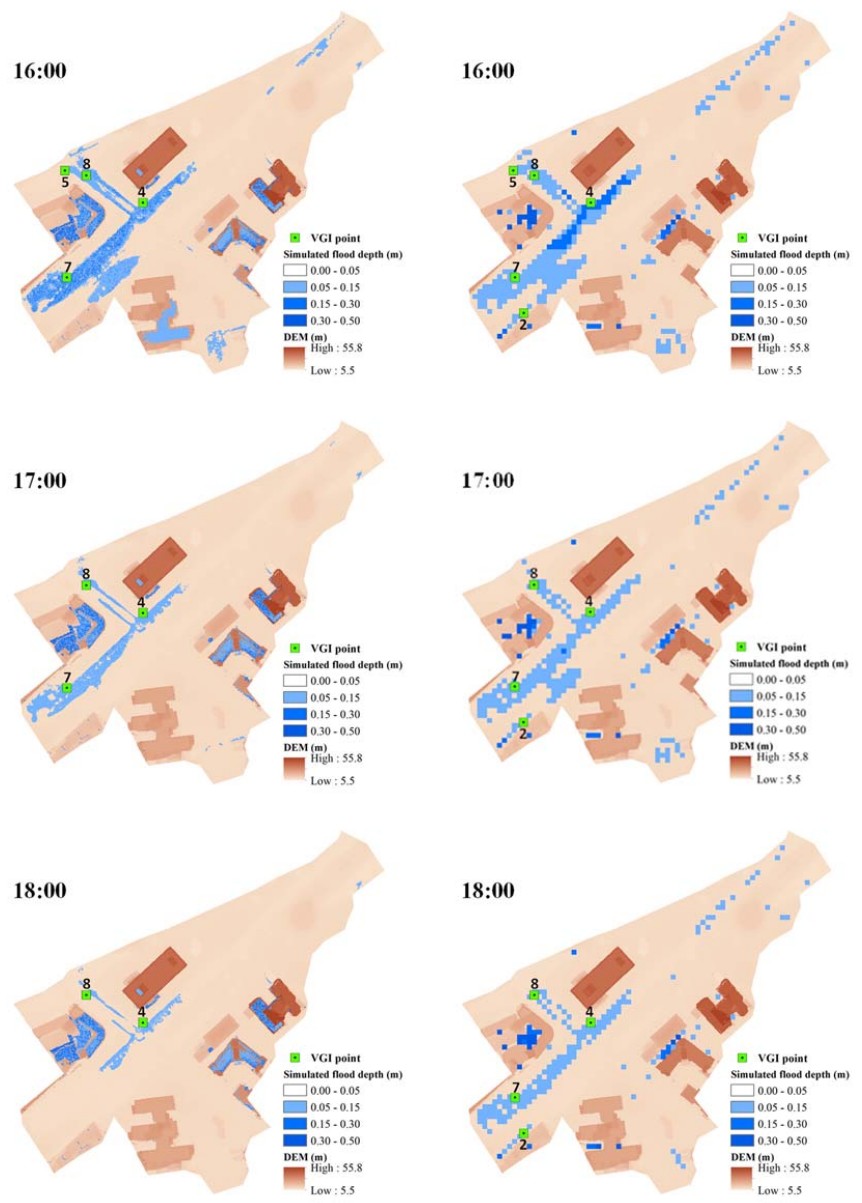

Figure 8: Simulated flood extents at different time under DEM resolution 0.5 m
(left) and DEM resolution 5 m (right).





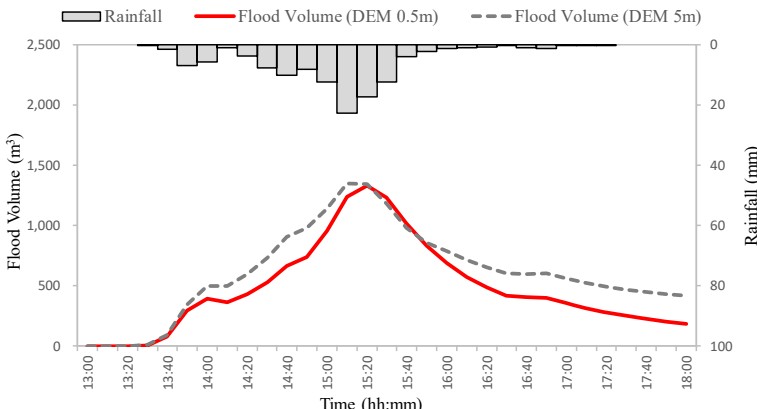

Figure 9: Comparison of simulated flood volume under two DEM resolutions.