# Peer review of "Using unmanned aerial vehicle and volunteered geographic information to sophisticate urban flood modelling"

_Hydrology and Earth System Sciences, 2020_

## Referee Comment (RC1) · Anonymous Referee #1 · 27 Mar 2020

General comments: The authors attempt to combine (1) UAV aerial surveying data with, (2) volunteered geographic information (VGI) and (3) computational flood simulation (CFS). Combining all three approaches is a useful topic and the authors are encouraged to pursue further fieldwork and research in this area. However, the paper skims the surface of each topic, has poor quality input data, buries the details of data analysis, incorporates a number of poor/dubious practices, and hides the quality of output data inside lumped categories. The conclusion of the paper that a higher resolution DEM produces better CFS results is common sense and hardly new. Other factors that are arguably more important are resolving critical sub grid scale features such as walls, and how these can be incorporated into a coarser (or variable resolution computation grid). The factors above and comments below make it impossible to recommend publication.

Specific comments:  c The paper covers a small spatial area and the limitations of UAV's in this regard is not discussed.  c Boundary conditions are the edge of the spatial domain are not considered/discussed.  c A freeway/motorway takes up a substantial proportion of the study domain, but is removed from the DEM without sufficient information on how the DEM was estimated where this was removed, or how roughness/friction parameters were estimated.  c Vegetation takes up a substantial proportion of the study domain, but is removed from the DEM without sufficient information on how the DEM was estimated where this was removed, or how roughness/friction parameters were estimated.  c The study only uses 3 ground control points for UAV surveys which is not enough. At least 8 required, with many studies recommending 16+.  c There is no discussion of flight regulations limiting UAV operations in urban areas and other similar considerations.  c The study talks about a computational sewer model being used, but provides no details of this and where sewers were or how flow was accounted for.  c The study provides very limited details of the CFS model. Other papers are referenced, but no local information is provided on roughness of different terrain types etc that must be used inside the CFS but are local to the study area.  c The paper provides irrelevant equations and information about DEM reconstruction and camera lens distortion (section 2.1). These are a red herring and completely irrelevant. The authors used Pix4D to do their aerial image processing and have not implemented the equations themselves. Pix4D or Agisoft Metashape are the appropriate software packages for this type of work, but the authors should spend more time discussing the appropriate workflow for data processing. It is likely that they did not follow a recommended workflow since they only used 3 ground control points.  c The timestamp of the photos from 'picture posting time' is not at all defensible. The authors should extract the EXIF information from the photos and look at image capture time. If images were captured with a cell phone then the timestamps should be accurate.  c The authors did not adequately survey flood depth at locations from

the VGI images. They should have gone out with an RTK GPS survey system and a ruler after the flood and measured the spatial location of depth reference points and the associated depth. Not doing this ('flood depth estimated from photos") is very poor practice. • Other errors throughout the paper from lack of attention to detail (see technical comments below) also call the accuracy and research quality of the paper into question. • Scaling of figures 7 and 8 is poorly selected and shows nothing of the fine scale DEM at ground level which is critical for the flood modelling. The selection of this scaling raises questions as to whether it was selected on prpose to hide a poor quality DEM at ground level. • Data in table 4 have been thresholded by the arbitrary category of water depth over 5 cm deep. This simple thresholding makes it far easier for data to appear correct (i.e. assigned to binary over/under categories). The data should compare actually flood depth (from ground truth measurements at VGI photo locations compared to observed water levels in photos) with flood simulation depth and quantify the error (discrepancy between the two). • The paper is well written in some sections, and poorly in others. Many sections would benefit from a rewrite, information being removed, information being added, or information being moved to other sections. This is beyond the scope of what is expected from a reviewer, hence I have only listed some of the obvious errors, suggestions, and grammatical corrections in the technical corrections below. Hopefully these will help the authors to rework the paper to become a high-quality conference paper, or with very thorough reworking and further analysis it may possible for it to be eventually published as a journal article. However, it may be faster for the authors to record another more thorough dataset (in a more suitable location) to analyse for a future journal paper.

See the attached file for the full review that includes technical corrections.

Feel free to pass on the attached review file to the authors. The review is detailed, tough, but hopefully fair and should improve the manuscript or a future repeat of the study. The authors should continue work in this area as it is important, however the quality of this publication is not quite up to international journal standards.

Please also note the supplement to this comment:
https://www.hydrol-earth-syst-sci-discuss.net/hess-2020-59/hess-2020-59-RC1-
supplement.pdf

———————————————————
59, 2020.

**Supplement:**

**Paper Details**

Title: Using unmanned aerial vehicle and volunteered geographic information to sophisticate urban flood modelling

Author(s): Yuan-Fong Su, Yan-Ting Lin, Jiun-Huei Jang, and Jen-Yu Han

MS No.: hess-2020-59

MS Type: Research article

Iteration: Initial Submission

Reviewed paper version: hess-2020-59.pdf

**Review Details**

**Recommendation:**

Reject manuscript. It would be more suitable for a conference presentation.

**General comments:**

The authors attempt to combine (1) UAV aerial surveying data with, (2) volunteered geographic information (VGI) and (3) computational flood simulation (CFS). Combining all three approaches is a useful topic and the authors are encouraged to pursue further fieldwork and research in this area. However, the paper skims the surface of each topic, has poor quality input data, buries the details of data analysis, incorporates a number of poor/dubious practices, and hides the quality of output data inside lumped categories. The conclusion of the paper that a higher resolution DEM produces better CFS results is common sense and hardly new. Other factors that are arguably more important are resolving critical sub grid scale features such as walls, and how these can be incorporated into a coarser (or variable resolution computation grid). The factors above and comments below make it impossible to recommend publication.

**Specific comments:**

- The paper covers a small spatial area and the limitations of UAV's in this regard is not discussed.
- Boundary conditions are the edge of the spatial domain are not considered/discussed.
- A freeway/motorway takes up a substantial proportion of the study domain, but is removed from the DEM without sufficient information on how the DEM was estimated where this was removed, or how roughness/friction parameters were estimated.
- Vegetation takes up a substantial proportion of the study domain, but is removed from the DEM without sufficient information on how the DEM was estimated where this was removed, or how roughness/friction parameters were estimated.
- The study only uses 3 ground control points for UAV surveys which is not enough. At least 8 required, with many studies recommending 16+.
- There is no discussion of flight regulations limiting UAV operations in urban areas and other similar considerations.
- The study talks about a computational sewer model being used, but provides no details of this and where sewers were or how flow was accounted for.

- The study provides very limited details of the CFS model. Other papers are referenced, but no local information is provided on roughness of different terrain types etc that must be used inside the CFS but are local to the study area.
- The paper provides irrelevant equations and information about DEM reconstruction and camera lens distortion (section 2.1). These are a red herring and completely irrelevant. The authors used Pix4D to do their aerial image processing and have not implemented the equations themselves. Pix4D or Agisoft Metashape are the appropriate software packages for this type of work, but the authors should spend more time discussing the appropriate workflow for data processing. It is likely that they did not follow a recommended workflow since they only used 3 ground control points.
- The timestamp of the photos from 'picture posting time' is not at all defensible. The authors should extract the EXIF information from the photos and look at image capture time. If images were captured with a cell phone then the timestamps should be accurate.
- The authors did not adequately survey flood depth at locations from the VGI images. They should have gone out with an RTK GPS survey system and a ruler after the flood and measured the spatial location of depth reference points and the associated depth. Not doing this ('flood depth estimated from photos") is very poor practice.
- Other errors throughout the paper from lack of attention to detail (see technical comments below) also call the accuracy and research quality of the paper into question.
- Scaling of figures 7 and 8 is poorly selected and shows nothing of the fine scale DEM at ground level which is critical for the flood modelling. The selection of this scaling raises questions as to whether it was selected on prpose to hide a poor quality DEM at ground level.
- Data in table 4 have been thresholded by the arbitrary category of water depth over 5 cm deep. This simple thresholding makes it far easier for data to appear correct (i.e. assigned to binary over/under categories). The data should compare actually flood depth (from ground truth measurements at VGI photo locations compared to observed water levels in photos) with flood simulation depth and quantify the error (discrepancy between the two).
- The paper is well written in some sections, and poorly in others. Many sections would benefit from a rewrite, information being removed, information being added, or information being moved to other sections. This is beyond the scope of what is expected from a reviewer, hence I have only listed some of the obvious errors, suggestions, and grammatical corrections in the technical corrections below. Hopefully these will help the authors to rework the paper to become a high-quality conference paper, or with very thorough reworking and further analysis it may possible for it to be eventually published as a journal article. However, it may be faster for the authors to record another more thorough dataset (in a more suitable location) to analyse for a future journal paper.

**Technical corrections:**

| Line number | Previous version | Correction |
|---|---|---|
| Title | Using unmanned aerial vehicle and volunteered geographic information to sophisticate urban flood modelling | Using an unmanned aerial vehicle and volunteered geographic information for sophisticated urban flood modelling |
| 15 | simulation (CFS) to reconstruct the flash flood event occurred in 14 June 2015, GongGuan, Taipei. | simulation (CFS) to reconstruct the flash flood event that occurred on the 14th of June 2015 in GongGuan, Taipei. |
| 17 | acquired from social network are served to establish and validate the CFS model, respectively. | acquired from social networks are used to establish and validate the CFS model. |
| 19 | The results show that flood scenario | The results show that the flood scenario |

| 26 | Flash flooding resulted from extreme heavy rainfall | Flash flooding resulted from extremely heavy rainfall |
|---|---|---|
| 47 | DEM data are derived by airborne Lidar | DEM data are derived from airborne Lidar |
| 50 | two raising techniques namely unmanned aerial vehicle | two rising techniques namely unmanned aerial vehicle |
| 53 | (DEM) derived by UAV have similar performances in urban | (DEM) derived from UAV aerial imagery have similar performance in urban |
| 58 | study of 2013 Boulder flood. | study of the 2013 Boulder flood. |
| 64 | The DEM generated by UAV can be served as the boundary conditions to increase the spatial resolution of CFS | Presumably this should be: "The DEM generated from UAV aerial imagery can be used as the boundary conditions to increase the spatial resolution of CFS"

However, I have no idea what they are talking about with 'boundary conditions to increase the spatial resolution of CFS'? DEM resolution is arbitrary and depends on how SfM or LIDAR data are resampled and output. Boundary conditions at the edges of the spatial extent of the computational domain should be properly addressed and this information is not clear in the paper. |
| 74 | rain gauge are shown in the Fig. 2. The DEM derived by UAV and the flood photos collected from VGI are served to establish and validate the CFS, respectively | rain gauge are shown in Fig. 2. The DEM derived from UAV aerial imagery and the flood photos collected from VGI are used to establish and validate the CFS. |
| 82-96 | | Remove this section. They do not independently implement this technique. They simply use Pix4D and the actual algorithms contained within are far more complex than the information provided in this section. Focus on the workflow for image processing in Pix4D. |
| 97 | DJI Phantom 2 Vision+ (Da-Jiang Innovations) which weights 1.2 kg and has a camera with 4384×2466 pixels. | DJI Phantom 2 Vision+ (Da-Jiang Innovations) which weighs 1.2 kg and has a camera with resolution of 4384×2466 pixels. |
| 105-108 | | 3x GCPs is not nearly enough! |
| 109-116 | | Remove the section on lens distortion. Completely irrelevant to the study. Again Pix4D calculates and accounts for lens distortion. They do not do it themselves. Remove table 2 about the camera on the UAV. It is irrelevant and does not generalise to the equipment used by other researchers. |
| 118 | The vegetation such as shrubs and grasses is detected by | Vegetation such as shrubs and grass were detected by |
| 117-125 | | I am dubious about their psudo NDVI method of vegetation detection from RGB imagery and the thresholding to detect the viaduct. How 'removed' elements were then accounted for is not stated. Interpolation how? What roughness values were assigned to the unknown terrain? How was water drainage accounted for on building roofs? Down |

| | | pipes etc? How were walls and other important aspects accounted for? |
|---|---|---|
| 130-134 | Based upon the Act, the VGI data used in this study are collected from the most famous Bulletin Board System (BBS) in Taiwan named PTT. There are 8 photos collected from PTT posted during 15:20~16:30 on 14 June 2015. From these photos, we visually identified 8 locations in the study area as shown in Fig. 6. The timestamp and the virtual water depths in these photos are served to validate the CFS model. Although the timestamp when photos were posted on internet may | Based upon the Act, the VGI data used in this study were collected from the most well-known Bulletin Board System (BBS) in Taiwan named PTT. There were 8 photos collected from PTT posted during 15:20~16:30 on 14 June 2015. From these photos, we visually identified 8 locations in the study area as shown in Fig. 6. The timestamp and the virtual water depths in these photos were used to validate the CFS model. Although the timestamp when photos were posted on the internet may |
| 135-137 | | Photo capture timestamps could be extracted from EXIF information stored within the image data. Most images have this info. Sometimes GPS data will also be contained in EXIF information. This should be checked. Flood depth estimation from photos is very poor practice. Field surveying after floods should be used to measure water depths corresponding to observations from photos. |
| 139-150 | | It is not clear where the sewer system is within the computational domain. It is also not clear how boundary conditions at the edges of the computation domain are accounted for (i.e. flow in and out of the domain). The sewer system will also connect out of the computational domain, the effects of which should be accounted for. |
| 153-157 | | Three GCPs are not enough! Agisoft recommends 10-15+ https://www.agisoft.com/index.php?id=34 More GCPs are needed if also used for independent validation of DEM and Orthomosaic spatial accuracy. |
| 159-161 | | This is methods not results. |
| 159 | DEM resolution on flood simulation, the gird meshes of the CFS | DEM resolution on flood simulation, the grid meshes of the CFS |
| 163 | in which the VGI points out of the 8 locations are marked if the simulated flood depths exceed 0.05 m. | in which the VGI points of the 8 locations are marked if the simulated flood depth exceeds 0.05 m. |
| 163-169 | | This >0.05 m depth criteria is completely arbitrary and is a way to divide the data into two lumped categories (flood vs no flood) which makes their results appear artificially better. They should compare simulated with measured depth directly and quantify the error properly. |
| 173 | "This implies that, when DEM resolution decreases, the topography becomes | Not really! How was sub grid scale roughness accounted for? Should this say: |

| | rugged, the friction increases, and the flood water travels slower." | "This implies that, when DEM resolution increases, the topography becomes rugged, the friction increases, and the simulated flood water travels slower." |
|---|---|---|
| 176-182 | The timestamps and estimated water depths (WD) are obtained from the VGI photos in Fig. 6, and the flood durations at the eight VGI points when the water depth exceeds 0.05 m are determined based on the CFS results. It is seen that the timestamps of VGI photos all lie within the simulated flood duration at the points with observed WD larger than 0.05 m (points #1, #2, #4, #7, and #8). At the rest points, the simulated and observed WDs are both smaller than 0.5 m. This good agreement between observation and simulation reveals that the flood model is accurate in rebuilding the process of flood transport under both DEM resolutions. | This arbitrary lumping into >0.05 m depth does not correspond to 'good agreement'. They should measure flood depths properly, not just estimate them, then quantify the error (predicted – observed).
There is also presumably a typo of "WDs are both smaller than 0.5 m" which likely should be "WDs are both smaller than 0.05 m". |
| 200 | For disaster emergency response in regional scale, flood simulation under coarse grid resolution is enough to gain a fast and overall understanding of flood pattern. | For disaster emergency response at regional scales, flood simulation under coarse grid resolution is enough to gain a fast and overall understanding of flood patterns. |
| 205 | CFS in urban area is a challenging | CFS in urban areas is a challenging |
| 206 | Aided by the rapid growing | Aided by the rapidly growing |
| 208 | we adopt the UAV and VGI to sophisticate CFS modeling in the reconstruction of a flash flood event occurred | we use UAV and VGI data for sophisticated CFS modeling to reconstruct a flash flood event that occurred |
| 215 | applicable in acquiring necessary data for high-resolution CFS. | applicable for acquiring the necessary data for high-resolution CFS. |
| Table 1 | | Sloppy typos. Possibly indicative of many more hidden errors.
"San Paulo" -> "São Paulo"
"Daintree, New Zealand" -> "Daintree, Australia" |
| Table 2 | | Irrelevant. All other researchers will have different cameras and don't care about the specific camera used. Just discuss the workflow for image processing in PIX4D where camera parameters were determined and imagery is de-warped before further processing. |
| Table 3 | | When generating a georeferenced orthomosaic or DEM from aerial imagery and Structure from Motion (SfM) techniques, more GCPs are needed for orthorectification and DEM generation than just 3 validation points. Yes, the UAV has a rough GPS location, but it is not RTK or PPK accuracy and should only be used for aligning images. Or if accurate DEMs are not required then at least discuss this. |

| | | | It is particularly critical for vertical elevations and generation of DEMs to use enough GCPs distributed throughout the study site. |
|---|---|---|---|
| Table 4 | | | This is not a 'comparison between CFS and VGI results'. This is arbitrary thresholding to make data correspondence look better. Just show predicted vs observed and quantify the difference! |
| Figure 3 | | | Their workflow doesn't make a lot of sense and doesn't follow the same sequence/layout as most other people who use Pix4D or Agisoft Metashape for SFM.

Also, how do they claim to use only 3 GCPs for 'Point cloud with absolute 3D coordinates', then at the next step also do 'Accuracy assessment'? Independent GCPs from those used for georeferencing are needed for accuracy assessment. |
| Figure 4 | | | This is irrelevant to the study. They have not independently implemented these algorithms, but are just using Pix4D, so no point showing any diagrams like this. |
| Figure 5 | | | Motorway takes up a large part of the DEM, as does vegetation. It is not at all clear how this is accounted for after it is 'removed'.
The 3 GCPs are not enough, nor are they properly distributed throughout the study domain.
There are unknown edge effects in the orthomosaic/DEM. Usually a UAV is set to fly a regular grid with zig-zag lines with 80% front overlap of images and 60% side overlap of images (more overlap is better). This then generates a DEM and orthomosaic where the edges are low accuracy (due to insufficient overlap), with edge areas being cropped out of the final orthomosaic and DEM. Here there is a strange scattering of points and rough boundaries at the edges of the orthomosaic which raises questions about the accuracy of the orthomosaic, DEM and the UAV flight paths used.
The orthomosaic and DEM are cropped in figure 6 and beyond (which is good), however the anomalies in figure 5 are not accounted for. |
| Figure 6 | | | Check EXIF information for photo capture time. This information may be scrubbed from images automatically by PTT, but is worth checking.
Photo locations should be surveyed with RTK GPS and depth measured with a ruler by comparing water level on reference objects such as walls, buildings, bike tires etc. |
| Figure 7 | | | The colour scheme and graduation does not resolve the finer scale features needed for CFS. It |

| | | |
|---|---|---|
| | | would be better with a logarithmic scale. Or just from 5-6 m and buildings will all be one colour. Potentially the colour scheme was selected to hide a poor quality underlying DEM. |
| Figure 8 | | Again poor selection of DEM scale. Lumped flood bins used rather than a continuous colour bar. Why? To hide problems? Or just poor choice of data representation? Where are the sewers and manholes? How are they accounted for? Why did they choose to run the study in an area where the motorway blocks so much of the computational domain? |
| Figure 9 | | Validation? Upstream flow into computational domain? Which is better? Results of 0.5m or 5m simulation? No real way to prove it as no external validation. The VGI data is hardly proof. Even if 0.5m grid is more accurate (as everyone expects) this is not news. Finer grid usually gives better computational results. |

**Further specific suggestions:**

| Section | Suggestion |
|---|---|
| Abstract | Quantify the accuracy, rather than saying 'more accurately'. |
| 1 Introduction | DEM resolution is important, but the proper representation of sub grid scale features is often more important (e.g. wall, stop-banks, culverts, bridges etc). How these are represented in a coarse DEM is critical. Multi resolution DEMs are possible. Also discuss how roughness is parameterised. I.e. if a modelling cell contains vegetation vs rocks vs concrete. This is also relevant at the end of the results section where it talks about computational efficiency and grid resolution. |
| 1 Introduction | Discusses DEMs from UAVs and LIDAR. See technical correction above about explicitly stating 'UAV aerial imagery'. LIDAR can also be flown on UAVs. |
| 4 Conclusions | 4 Summary and conclusions |

---

## Referee Comment (RC2) · Anonymous Referee #2 · 10 Apr 2020

**1 General comments**

The authors present a case study where UAV and VGI photos are used to run a CFS including a coupling between surface and sewer system flows and they evaluate the influence of the DEM resolution on simulation results. The UAV images are processed to create a high resolution DEM. Results of the numerical simulation are compared to VGI photos in terms of water depth to assess its correctness. A comparison is made between 2 DEM resolutions, and the conclusion is that higher resolution yields better results. The ideas developed in this paper are interesting, and are a useful contribution towards simple and flexible hydraulic numerical simulation by using remote sensing and

crowdsourcing. Nevertheless, the paper has to be improved in many points. There is a global lack of scientific rigor, of precision. Many important details are not mentioned, whereas some parts are out of the scope of the paper. Furthermore, the overall English quality is poor. Because of all the weaknesses, both on form and content, I suggest this paper should not be accepted for publication.

**2 Specific comments**

Many remarks are written in the technical corrections part, but in this part I sum up comments on three major issues:

**2.1 DEM generation**

This section should be totally rewritten. General considerations about DEM generation and application to the case study are mixed, and not put in a correct and logical order. Camera calibration should be addressed before absolute positioning of images. The authors should consider being shorter on generalities and give more information about their own input during this step. The authors do not say how they replace the DEM parts they remove: motorway bridge and vegetation. How is the new altitude chosen? Don't bridge pillars or vegetation have an influence on the surface flow? How do the authors apply the method from Rabatel et al. without removing the NIR filter of the camera in the first place?

**2.2 CFS model**

Very little information about the model is given in the paper. The reader is referred to 2 other papers (Jang et al. 2018 & 2019). The effort made not to be redundant

is appreciated, but some more details are needed about: 1) the numerical coupling 2) the manholes location and implementation in the model 3) the chosen hydraulic parameters, roughness distribution, boundary conditions... 4) the time extent of the simulation

**2.3 Results**

The DEM accuracy is checked on the same points that were used to do its absolute georeferencing, which does not prove anything. The authors' assertion that coarser resolution leads to lower quality results is not confirmed by the presented arguments. Water depths of both simulations are compatible with all the VGI photos. Water depths on rooftops are not measured nor observed (or some information is missing in the paper). Flood duration differs between resolutions, but none is contradicted by some observations. The conclusion concerning the DEM resolution is not supported enough to be stated as it is.

**3 Technical corrections**

| Location | Comment |
| --- | --- |
| title | . . . to **sophisticated urban** |
| l.15 | . . . flash flood event **that occurred on** |
| l.17 | . . . network are **used to establish** |
| l.18 | . . . data **is resampled** |
| l.21 | . . . and VGI **lowers the** |
| l.26 | Flash **floods resulting from extreme rainfall** |
| l.28-29 & Tab 1 | Table 1: What is the point of this table? |
| | Is it really needed in this paper? |
| | Line 29 mentions life losses, but no figures about it in the table. |
| | The authors should consider deleting this table and modifying |
| | the text accordingly |
| l.52-53 | Citations only refer to VGI, not UAV techniques |
| l.63 | . . . can be **used as** |
| l.63-64 | Does "boundary conditions" mean "hydraulic boundary |
| | conditions"? More explanations are needed here |
| | to understand the link between boundary conditions |
| | and spatial resolution |
| l.69 | . . . event **that occurred on** |
| l.70 | . . . occurred between 13:00 **and 18:00** |
| l.71 | . . . 131.5 mm/h **from 14:30 to 15:30** |
| l.72 | . . . rainfall intensity **exceeded** |
| l.73 | . . . Chanxing street **near the National** |
| l.75 | VGI are **used to establish** |
| l.79 | "finally, the simulated results are compared" Contradiction with fig. 3 |
| | where VGI seems to be an input to the CFS. |

| | |
|---|---|
| Section 2.1 | This section should be totally rewritten. |
| DEM generated by UAV | General considerations about DEM generation and application to the case study are mixed and not put in a correct and logical order. Camera calibration should be addressed before absolute positioning of images. The authors should consider being shorter on generalities and give more information about their own input during this step |
| l.82 | . . . left side of **Fig. 3** |
| l.82-83 | . . . methods **for generating a DEM** |
| l.84 | . . . 2004). **They are based** |
| l.86, 92, 93 | . . . **coordinates** |
| l.90 | The six parameters [. . .] are **determined during exterior orientation** |
| l.96 | What resolution is chosen for resampling? |
| l.100 | . . . condition **from 06:00** |
| l.117 | Vegetation is removed: how the new ground altitude is computed? With which roughness? Does vegetation have no impact on the flow? How the results are biased by this removal? |
| l.117 | The viaduct is removed: how the new groud altitude is computed? With which roughness? Do the viaduct's pillars have no impact on the flow? |
| l.118 | . . . shrubs and **grass** |
| l.120-121 | The authors did apparently not fully understand the publication by Rabatel et al., which needed a removal of the NIR blocking filter inside the camera. This paper does not mention this removal, nor gives any detail about the specific linear combination. |
| l.121-122 | Some of the surrounding buildings must have heights above the 9 m threshold. However the authors claim that they remove the viaduct, but not the buildings (l. 124-125) |
| l.124 | . . . **flow smoothly on ground surface** |

| | |
|---|---|
| l.125 | Transverse is not the adequate word |
| l.125 | There is no mention of (parked) cars and street furniture. They might have an influence on the flow. Are they in the DEM or not? The authors should give a comment on this. |
| l.133 | . . . these photos are **used to** |
| l.135 | The described hydrological event lasts for a few dozens of minutes. A slight shift in the image timestamp could lead to totally wrong information. The authors should only take into account photos for which the timestamp corresponds to the moment when it was taken, not when it was posted on a social network. |
| l.135-136 | The authors should give an idea of uncertainties yielded by manual flow depth estimation. What about an automatic water level estimation? |
| Section 2.3 CFS model | This section lacks details about the chosen hydraulic parameters (roughness, boundary conditions) for both OFM and SFM. It is not clear how the manhole positions are determined: DEM and water levels are obtained from images, why not the manholes? It is also not clear which type of interaction between OFM and SFM is applied. The authors should also give the time limits of their simulations. |
| l. 153 | The DEM accuracy is checked on 3 GCPs. The same 3 GCPs were used to perform the DEM georeferencing (see lines 104-108). The authors should use a different set of control points than those who were used to process the data, with a better spatial distribution in the modeled area. Furthermore, 2 DEM are created, with distinct resolutions. Accuracy should be checked for both. |
| l. 156 | The authors did not give the initial DEM resolution |
| l. 159 | . . . simulation, the **grid meshes** |

| l. 167-169 | The authors state that water on rooftops is better simulated by the fine resolution model. Could they give some validation criteria to explain this statement? Could they explain why water should be on rooftops? Does their model take water evacuation from rooftops into account? |
|---|---|
| l.173 | Higher: not much higher. The authors should give the values. |
| l.174 | Lower resolution implies that small terrain features are not represented, so the topography should be smoother. The authors observe the opposite. The authors should check the DEM resampling, and give more comments about their observation. |
| l.178 | . . . results. It **can be seen** |
| l.180 | . . . At the **remaining points, the simulated** |
| l.184-186 | I can not see the causal relationship between the fact that there are no VGI photos after 15:40 and the overestimation of flood duration. Since there are no photos, it is not possible to conclude in a way or another. |
| l.195 | . . . than **that that**. . .**.** |
| l.207-208 | DEM updates, and resulting CFS are not in real time, since weather conditions generally prevent UAVs to fly during or just after heavy rainfalls. |
| l.211 | It has not been demonstrated in the paper that fine resolution modeling results are better. The comparison presented in table 4 shows that both resolutions give results consistent with VGI photos. Other validation data is needed to be able to draw this conclusion on this case study. |
| Table 1 | What is the point? The list of flash floods that occurred in 2019 is not relevant to the paper. Moreover, events after July, 8th are not listed. |

| | |
|---|---|
| Table 3 | The accuracy is checked on the very same points used in the DEM production process. Low error values are thus expected. It gives no information on the accuracy of the DEM everywhere else. |
| Figure 1 | Caption: rain gauge (as in the text) |
| | In the bar chart, times are written above bars, I guess they should lie between bars |
| Figure 2 | Rain gauge (as in the text) |
| | Top left image is not necessary. I do not understand the gray levels in top right image : shade or altitude? |
| | Bottom image is very dark, authors should improve brightness and contrast. |
| | The cartographic scale is indicated only for the bottom image |
| Figure 3 | The right part is not consistent with the paper. VGI seems to be an input of CFS. Parameter of Drainage system: is there only one parameter? This parameter is not even mentioned in the paper. |
| Figure 4 | Not necessary in this paper |
| Figure 5 | - The caption reads: "images taken", but they must have undergone some processing. The authors should tell readers what is exactly displayed in fig. 5 |
| | - The image is distorted compared to the area shown in the following figures (stretched along Y axis). Why? |
| | - GCP #1 is located just next to the viaduct, so there is a discontinuity in altitude just nearby. It is recommended to select clear zones for GCPs |
| Figure 6 | All minutes in timestamps seem to be rounded to the nearest 10. Why? How water depths could possibly be estimated from photo #1? |

| Figure 7 | The choice of the color scale is not adequate to allow a good perception of low value altitudes (everything is blue). Moreover, the rainbow color scale should be avoided: https://www.nature.com/articles/519291d https://www.climate-lab-book.ac.uk/2014/end-of-the-rainbow/ |
| Figure 8 | Water depths between 0.00 and 0.05 m should appear in white according to the caption. I can not see any white pixel. Why? The difference between the colors corresponding to 0.15-0.30 and 0.30-0.50 water depths is too small to be distinguished. Water accumulates on roofs, especially for the fine resolution model. It seems very unlikely, especially for the building lying in the left. |

[Figure]

---

## Author Comment (AC1) · 26 May 2020

**Responses to Anonymous Referee #1**

**MS No.: hess-2020-59**

**General comments:**

The authors attempt to combine (1) UAV aerial surveying data with, (2) volunteered geographic information (VGI) and (3) computational flood simulation (CFS). Combining all three approaches is a useful topic and the authors are encouraged to pursue further fieldwork and research in this area. However, the paper skims the surface of each topic, has poor quality input data, buries the details of data analysis, incorporates a number of poor/dubious practices, and hides the quality of output data inside lumped categories. The conclusion of the paper that a higher resolution DEM produces better CFS results is common sense and hardly new. Other factors that are arguably more important are resolving critical sub grid scale features such as walls, and how these can be incorporated into a coarser (or variable resolution computation grid). The factors above and comments below make it impossible to recommend publication.

**Response:** Thank you for the comments. We made several changes on the validations and added some materials to make it clearer. Please see the following point-by-point responses.

**Specific comments:**

1  The paper covers a small spatial area and the limitations of UAV's in this regard is not discussed.

   **Response**: Compared with other surveying and mapping methods, UAVs are more easily deployed to quickly update the 3D spatial information after disasters. However, the flight height of UAVs is limited by the regulations in urban areas. Thus, in consideration of the limitation of flight height and the ground resolution requirements in the study area, the UAV was set to fly at a height of 100 meters to perform vertical surface shooting. Relevant discussions have been added to the revised manuscript.

2  Boundary conditions are the edge of the spatial domain are not considered/discussed.

   **Response**: For the CFS, the rainfall and DEM data are given at each grid center within the simulation domain. At the edge of the domain boundary, the water is allowed to outflow freely. The descriptions above are added to the revised manuscript.

3  A freeway/motorway takes up a substantial proportion of the study domain, but is removed from the DEM without sufficient information on how the DEM was estimated where this was removed, or how roughness/friction parameters were estimated.

   **Response**: The freeway is elevated and supported by the pillars at the centerline of the Keelung Road (shown in the following photo). The freeway is removed for the CFS because flood water is allowed to flow across underneath the viaduct. Since the ground elevations were observed by the UAV at the divisions between the viaduct lanes and the buildings on the roadsides, the DEM underneath the conduct can be estimated. For the CFS, the friction parameters are estimated by the Manning's coefficients subject to land covers (Chow, V.T., 1959). The discussions above have been added to the revised manuscript.

   **Reference:** Chow, V.T. (1959) Open Channel Hydraulics. McGraw-Hill, New York.

[Figure]

Fig. 1 The freeway is elevated and supported by the pillars at the centerline of the Keelung Road

4    Vegetation takes up a substantial proportion of the study domain, but is removed from the DEM without sufficient information on how the DEM was estimated where this was removed, or how roughness/friction parameters were estimated.

**Response:** The regions of vegetation are detected using the ExG-ExR binary index (Meyer and Neto, 2008) by subtracting the alternate excess red vegetative index ($ExR = 1.4r-b$) from the excess green vegetation index ($ExG = 2g-r-b$), where $r$, $g$, and $b$ are the chromatic coordinates. To consider the friction effects by the roughness of vegetation, the Manning's coefficient is set as 0.04 for the CFS.

**Reference:** Meyer, G. E. and Neto, J. C.: Verification of color vegetation indices for automated crop imaging applications. Computers and electronics in agriculture, 63(2), 282-293. https://doi.org/10.1016/j.compag.2008.03.009, 2008.

5    The study only uses 3 ground control points for UAV surveys which is not enough. At least 8 required, with many studies recommending 16+.

**Response**: The coordinates of the three GCPs were obtained by referring to the publicly released values of Taipei City Government and using the static positioning of Global Navigation Satellite System (GNSS) with positional accuracy in centimeter level. The difference between the coordinates obtained by these two methods can be used to evaluate the accuracy of the ground control points. The 0.5m and 5m DEMs are created and validated according to the initial UAV-based DEM with resolution of 0.03m. We have added more discussion on the GCP requirement in the revised manuscript. The reasons we used only three GCPs are (1) the study area is relative small (0.0637 km2) and the GPS information on the UAV could produce 3D coordinate with certain degree of accuracy; (2) there are exactly three GCPs released by the Taipei City Government in this study area and we also double check the released values with the static positioning of Global Navigation Satellite System (GNSS) with positional accuracy in centimeter level; (3) The number of GCP depends on the surveying areas, flight altitudes, resolutions and application goals. According to the user manual of Pix4D (https://support.pix4d.com/hc/en-us/articles/204272989-Offline-Getting-Started-and-Manual-pdf), a minimum number of 3 GCPs is required. To provide more information to readers, we have added more discussion on the GCP requirement in the revised manuscript.

6    There is no discussion of flight regulations limiting UAV operations in urban areas and other similar  considerations.

**Response**: In the study area, the flight height is limited under 100 meters according to the UAV operation regulations in urban areas. This statement has been added to the revised manuscript.

7    The study talks about a computational sewer model being used, but provides no details of this and  where sewers were or how flow was accounted for.

**Response**:  The pipelines and manholes of the sewer systems are displayed in the following Figure:

[Figure]

Fig. 2 The pipeline and manholes of the sewer systems in the study area.

When rain drops, the overland flow model (OFM) is firstly initiated for surface water routing. Then, the surface runoff travels for a distance and enters the sewer pipes via the street inlets to trigger the sewer flow model (SFM). When the sewer pipes get full, the sewer water surcharges back onto ground surface via the manholes or inlets. In the simulation process, the water exchanged between the two models are determined by weir and orifice functions via one-to-one relationship as shown in the schematic diagram below. These explanation have been added to the revised manuscript.

[Figure]

Fig. 3 Illustration of the water simulation process between ground surface, inlet, sewer pipe and manhole.

8    The study provides very limited details of the CFS model. Other papers are referenced, but no local  information is provided on roughness of different terrain types etc that must be used inside the CFS  but are local to the study area.

**Response**: The Manning's coefficient n is used to represent the surface roughness subject to land covers. Since the land covers are mostly concrete and short grass, the value of n slightly varies from 0.03 to 0.04 according to Chow, V.T. (1959). Although the skin friction represented by Manning's n changes little, the form frictions caused by road curbs and building walls can be more significantly presented as DEM resolution increases. The discussions above and more details about the CFS, as shown in the previous comments, have been added to the revised manuscript.

9    The paper provides irrelevant equations and information about DEM reconstruction and camera lens  distortion (section 2.1). These are a red herring and completely irrelevant. The authors used Pix4D to  do their aerial image processing and have not implemented the equations  themselves. Pix4D  or  Agisoft Metashape are the appropriate software packages for this type of work, but the authors should  spend more time discussing the appropriate workflow for data processing. It is likely that they did not  follow a recommended workflow since they only used 3 ground control points.

**Response**: We revised the Section 2.1 by adding necessary information about the determination of roughness and DEM for areas of vegetation and underneath the viaduct. The Pix4D workflow of data processing is displayed in the following Fig. 4 and is identical to the manual suggestions (Pix4D, 2017). The basic theories of collinearity and lens distortion are keep in the manuscript for reader's reference. According to the Pix4D manual, three GCPs have met the basic requirement for UAV image processing and validation. We have added some discussion on the GCP requirement in the revised manuscript.

Pix4D, User Manual v4.1, pp.26. https://support.pix4d.com/hc/en-us/articles/204272989-Offline-Getting-Started-and-Manual-pdf. 2017

[Figure]

Fig. 4 Conceptual flowchart of this study (the VGI photo was adopted from PTT, Taiwan)

10 The timestamp of the photos from 'picture posting time' is not at all defensible. The authors should extract the EXIF information from the photos and look at image capture time. If images were captured with a cell phone then the timestamps should be accurate.

**Response**: The VGI photos acquired from internet, for example from the PTT in this study, are usually not the original photos and therefore the EXIF information is not available. We checked it using http://metapicz.com. However, we are appreciate for this suggestion and relevant explanation is added to the revised manuscript.

11 The authors did not adequately survey flood depth at locations from the VGI images. They should have gone out with an RTK GPS survey system and a ruler after the flood and measured the spatial location of depth reference points and the associated depth. Not doing this ('flood depth estimated from photos") is very poor practice.

**Response**: Our estimation of flood depth form photos are based on some obvious targets such as wheel size of bikes and height of road curbs. The geometry of these targets are standard so that the flood depth can be estimated by mutual comparison.

12 Other errors throughout the paper from lack of attention to detail (see technical comments below) also call the accuracy and research quality of the paper into question.

**Response**: All the technical comments have been reviewed and the corresponding revision have been made (see the responses to technical corrections for detail).

13 Scaling of figures 7 and 8 is poorly selected and shows nothing of the fine scale DEM at ground level which is critical for the flood modelling. The selection of this scaling raises questions as to whether it was selected on purpose to hide a poor quality DEM at ground level.

**Response**: The scaling of the two figures are based on a continuous classification of DEM automatically generated by a GIS software. The ground levels are not displayed in detail because the height of buildings outweighs the variation of ground levels which results in the stretch of color bar. In order to highlight the details of ground level, the scales of these two figures have been adjusted in the revised manuscript and are displayed as below:

[Figure]

Fig. 5 The processed (a) orthophoto and the DEMs with spatial resolution of (b) 0.5 m (c) 5 m

[Figure]

[Figure]

Fig. 6 Simulated flood extents at different time under DEM resolution of 0.5 m (left) and DEM resolution of 5 m (right).

14  Data in Table 4 have been thresholded by the arbitrary category of water depth over 5 cm deep. This  simple thresholding makes it far easier for data to appear correct (i.e. assigned to binary over/under  categories). The data should compare actually flood depth (from ground

truth measurements at VGI photo locations compared to observed water levels in photos) with flood simulation depth and quantify the error (discrepancy between the two).

**Response**: For flood impact assessment, binary scaling of flood depth is commonly used because certain water depths have specific meanings. For example, 0.05m represents the height of ankle, when water depth exceeds it, people experience inconvenience; 0.3m is the depth above which furniture damages start to take place; 0.5m of water depth is the lower bound for compensation application in Taiwan. Therefore, the scalings in Table 4 and Figure 8 are not randomly selected but deliberately arranged to highlight the impacts of flooding.

As to the comparison with observed water level, the lack of onsite measured data is always the issue for CFS validation. This study proposes an approach to extract useful information by image processing technologies from VGI and UAV photos for urban flood modeling. From the comparison of CFS results with VGI photos, the building sidewalls and terrain depressions are demonstrated to have a great influence on flood extent, depth, and occurrence which can only be simulated by the CFS with high-resolution DEM. The original statements have been revised according to the above discussions.

15    The paper is well written in some sections, and poorly in others. Many sections would benefit from a rewrite, information being removed, information being added, or information being moved to other sections. This is beyond the scope of what is expected from a reviewer, hence I have only listed some of the obvious errors, suggestions, and grammatical corrections in the technical corrections below. Hopefully these will help the authors to rework the paper to become a high-quality conference paper, or with very thorough reworking and further analysis it may possible for it to be eventually published as a journal article. However, it may be faster for the authors to record another more thorough dataset (in a more suitable location) to analyse for a future journal paper.

**Response**: The original manuscript has been thoroughly revised by adding necessary information, removing unnecessary parts, arranging the text structures, and correcting the errors in grammar and methodology according to the suggestions by the reviewers. Hopefully, these revisions will make this paper more acceptable for publication in HESS.

**Technical corrections:**

| Line | Previous version | Correction | Response |
|------|-----------------|------------|----------|
| Title | Using unmanned aerial vehicle and volunteered geographic information to sophisticate urban flood modelling | Using an unmanned aerial vehicle and volunteered geographic information for sophisticated urban flood modelling | Revised accordingly. |
| 15 | simulation (CFS) to reconstruct the flash flood event occurred in 14 June 2015, GongGuan, Taipei. | simulation (CFS) to reconstruct the flash flood event that occurred on the 14$^{th}$ of June 2015 in GongGuan, Taipei. | Revised accordingly. |
| 17 | acquired from social network are served to establish and validate the CFS model, | acquired from social networks are used to establish and validate the CFS model. | Revised accordingly. |

| | | | |
|---|---|---|---|
| | respectively. | | |
| 19 | The results show that flood scenario | The results show that the flood scenario | Revised accordingly. |
| 26 | Flash flooding resulted from extreme heavy rainfall | Flash flooding resulted from extremely heavy rainfall | Revised accordingly. |
| 47 | DEM data are derived by airborne Lidar | DEM data are derived from airborne Lidar | Revised accordingly. |
| 50 | two raising techniques namely unmanned aerial vehicle | two rising techniques namely unmanned aerial vehicle | Revised accordingly. |
| 53 | (DEM) derived by UAV have similar performances in urban | (DEM) derived from UAV aerial imagery have similar performance in urban | Revised accordingly. |
| 58 | study of 2013 Boulder flood. | study of the 2013 Boulder flood. | Revised accordingly. |
| 64 | The DEM generated by UAV can be served as the boundary conditions to increase the spatial resolution of CFS | Presumably this should be: "The DEM generated from UAV aerial imagery can be used as the boundary conditions to increase the spatial resolution of CFS"

However, I have no idea what they are talking about with 'boundary conditions to increase the spatial resolution of CFS'? DEM resolution is arbitrary and depends on how SfM or LIDAR data are resampled and output. Boundary conditions at the edges of the spatial extent of the computational domain should be properly addressed and this information is not clear in the paper. | The original statements is not clear and has been revised as "The DEM generated from UAV provides detail terrain of an urban area which significantly increases the spatial resolution of CFS compared to traditional practices" Because the ground levels are given in grid unit in CFS, there exists an invisible wall between two adjacent grids with different elevations. When DEM resolution varies, the heights of these walls vary as well that affects the inter-cell water communications |
| 74 | rain gauge are shown in the Fig. 2. The DEM derived by UAV and the flood photos collected from VGI are served to establish and validate the CFS, respectively | rain gauge are shown in Fig. 2. The DEM derived from UAV aerial imagery and the flood photos collected from VGI are used to establish and validate the CFS. | Revised accordingly. |
| 82-96 | | Remove this section. They do not independently implement this technique. | We revised the sentences and mentioned the process is based on the Pix4D but the basic theory of |

| | | They simply use Pix4D and the actual algorithms contained within are far more complex than the information provided in this section. Focus on the workflow for image processing in Pix4D. | the collinearity condition are keep in the manuscript for readers' reference. |
|---|---|---|---|
| 97 | DJI Phantom 2 Vision+ (Da-Jiang Innovations) which weights 1.2 kg and has a camera with 4384×2466 pixels. | DJI Phantom 2 Vision+ (Da-Jiang Innovations) which weighs 1.2 kg and has a camera with resolution of 4384×2466 pixels. | Revised accordingly. |
| 105-108 | | 3x GCPs is not nearly enough! | According to the Pix4D manual, three GCPs have met the basic requirement for DEM processing (Pix4D, 2017). We have added more discussion on the GCP requirement in the revised manuscript. |
| 109-116 | | Remove the section on lens distortion. Completely irrelevant to the study. Again Pix4D calculates and accounts for lens distortion. They do not do it themselves. Remove table 2 about the camera on the UAV. It is irrelevant and does not generalise to the equipment used by other researchers. | We revised the sentences and mentioned the process is based on the Pix4D but the basic theory of the lens distortion are keep in the manuscript for readers' reference. |
| 118 | The vegetation such as shrubs and grasses is detected by | Vegetation such as shrubs and grass were detected by | Revised accordingly. |
| 117-125 | | I am dubious about their psudo NDVI method of vegetation detection from RGB imagery and the thresholding to detect the viaduct. How 'removed' elements were then accounted for is not stated. Interpolation how? What roughness values were assigned to the unknown terrain? How was water drainage accounted for on building roofs? Down pipes | The regions of vegetation are detected using the ExG-ExR binary index (Meyer and Neto, 2008) by subtracting the alternate excess red vegetative index (ExR = 1.4r−b) from the excess green vegetation index (ExG = 2g−r−b), where r, g, and b are the chromatic coordinates. To consider the friction effects by the roughness of vegetation, the Manning's coefficient is set as 0.04 for CFS. The water accumulated on rooftops because |

| | | etc? How were walls and other important aspects accounted for? | there are usually parapet walls with about 1 meter height on the rooftops around the borders of buildings in Taiwan. When the DEM resolution is high enough, the elevations of parapet walls can be represented by the grid-based mesh system in CFS and the water detention on the rooftops can be simulated. The rougher the grid/DEM resolutions, the faster the stored water will evacuate through the gaps between two adjacent grid cells. |
|---|---|---|---|
| 130-134 | Based upon the Act, the VGI data used in this study are collected from the most famous Bulletin Board System (BBS) in Taiwan named PTT. There are 8 photos collected from PTT posted during 15:20~16:30 on 14 June 2015. From these photos, we visually identified 8 locations in the study area as shown in Fig. 6. The timestamp and the virtual water depths in these photos are served to validate the CFS model. Although the timestamp when photos were posted on internet may | Based upon the Act, the VGI data used in this study were collected from the most well-known Bulletin Board System (BBS) in Taiwan named PTT. There were 8 photos collected from PTT posted during 15:20~16:30 on 14 June 2015. From these photos, we visually identified 8 locations in the study area as shown in Fig. 6. The timestamp and the virtual water depths in these photos were used to validate the CFS model. Although the timestamp when photos were posted on the internet may | Revised accordingly. |
| 135-137 | | Photo capture timestamps could be extracted from EXIF information stored within the image data. Most images have this info. Sometimes GPS data will also be contained in EXIF information. This should be checked. Flood depth estimation from photos is very poor | The photos acquired from PTT were not the original photos and the EXIF information were not available. We checked with http://metapicz.com/ |

| | | practice. Field surveying after floods should be used to measure water depths corresponding to observations from photos. | |
|---|---|---|---|
| 139-150 | | It is not clear where the sewer system is within the computational domain. It is also not clear how boundary conditions at the edges of the computation domain are accounted for (i.e. flow in and out of the domain). The sewer system will also connect out of the computational domain, the effects of which should be accounted for. | The sewer system is displayed in the response to specific comments #7. The surface water and pipe flow are allowed to flow freely at the edges of the computation domain. |
| 153-157 | | Three GCPs are not enough! Agisoft recommends 10-15+ https://www.agisoft.com/index.php?id=34 More GCPs are needed if also used for independent validation of DEM and Orthomosaic spatial accuracy. | This recommendation is for another software "PhotoScan", not the one "Pix4D" used in this study. According to Pix4D's manual, three GCPs are enough. We have added more discussion on the GCP requirement. |
| 159-161 | | This is methods not results. | The sentence "to discover the influence of DEM…" has been moved to method section 2.3. |
| 159 | DEM resolution on flood simulation, the gird meshes of the CFS | DEM resolution on flood simulation, the grid m e s h e s of the CFS | "grid" has been corrected. |
| 163 | in which the VGI points out of the 8 locations are marked if the simulated flood depths exceed 0.05 m. | in which the VGI points of the 8 locations are marked if the simulated flood depth exceeds 0.05 m. | Revised accordingly. |
| 163-169 | | This >0.05 m depth criteria is completely arbitrary and is a way to divide the data into two lumped categories (flood vs no flood) which makes their results appear artificially better. They should compare simulated with measured depth directly and quantify the error properly. | See the responses to specific comment #14 for detail. |

| 173 | "This implies that, when DEM resolution decreases, the topography becomes rugged, the friction increases, and the flood water travels slower." | Not really! How was sub grid scale roughness accounted for? Should this say: "This implies that, when DEM resolution increases, the topography becomes rugged, the friction increases, and the simulated flood water travels slower." | Because the ground levels are given in grid unit in CFS, there exists an invisible wall between two adjacent grids with different elevations. When DEM resolutions decrease, these walls become higher which result in larger blocking effects that reduce inter-cell water communications. This phenomenon explains why the flood water travels slower in the simulated results. |
|---|---|---|---|
| 176-182 | The timestamps and estimated water depths (WD) are obtained from the VGI photos in Fig. 6, and the flood durations at the eight VGI points when the water depth exceeds 0.05 m are determined based on the CFS results. It is seen that the timestamps of VGI photos all lie within the simulated flood duration at the points with observed WD larger than 0.05 m (points #1, #2, #4, #7, and #8). At the rest points, the simulated and observed WDs are both smaller than 0.5 m. This good agreement between observation and simulation reveals that the flood model is accurate in rebuilding the process of flood transport under both DEM resolutions. | This arbitrary lumping into >0.05 m depth does not correspond to 'good agreement'. They should measure flood depths properly, not just estimate them, then quantify the error (predicted – observed). There is also presumably a typo of "WDs are both smaller than 0.5 m" which likely should be "WDs are both smaller than 0.05 m". | The typo has been corrected. |
| 200 | For disaster emergency response in regional scale, flood simulation under coarse grid resolution is enough to gain a fast and overall | For disaster emergency response at regional scales, flood simulation under coarse grid resolution is enough to gain a fast and overall understanding of flood patterns. | Revised accordingly. |

| | | | |
|---|---|---|---|
| | understanding of flood pattern. | | |
| 205 | CFS in urban area is a challenging | CFS in urban areas is a challenging | Revised accordingly. |
| 206 | Aided by the rapid growing | Aided by the rapidly growing | Revised accordingly. |
| 208 | we adopt the UAV and VGI to sophisticate CFS modeling in the reconstruction of a flash flood event occurred | we use UAV and VGI data for sophisticated CFS modeling to reconstruct a flash flood event that occurred | Revised accordingly. |
| 215 | applicable in acquiring necessary data for high-resolution CFS. | applicable for acquiring the necessary data for high-resolution CFS. | Revised accordingly. |
| Table 1 | | Sloppy typos. Possibly indicative of many more hidden errors. "San Paulo" -> "São Paulo" "Daintree, New Zealand" -> "Daintree, Australia" | Table 1 has been deleted. |
| Table 2 | | Irrelevant. All other researchers will have different cameras and don't care about the specific camera used. Just discuss the workflow for image processing in PIX4D where camera parameters were determined and imagery is de-warped before further processing. | Although other researchers will have different cameras but we believe that this information is fundamental information for similar applications that should pay proper attention. |
| Table 3 | | When generating a georeferenced orthomosaic or DEM from aerial imagery and Structure from Motion (SfM) techniques, more GCPs are needed for orthorectification and DEM generation than just 3 validation points. Yes, the UAV has a rough GPS location, but it is not RTK or PPK accuracy and should only be used for aligning images. Or if accurate DEMs are not required then at least discuss this. It is particularly critical for vertical elevations and | The coordinates of the three GCPs were obtained by referring to the publicly released values by Taipei City Government and We further used the static positioning of Global Navigation Satellite System (GNSS) with positional accuracy in centimeter level to double check these values. The difference between the coordinates obtained by these two methods can be used to evaluate the accuracy of the ground control points. We have added more discussion on the GCP requirement in the revised manuscript . |

| | | | |
|---|---|---|---|
| | | generation of DEMs to use enough GCPs distributed throughout the study site. | |
| Table 4 | | This is not a 'comparison between CFS and VGI results'. This is arbitrary thresholding to make data correspondence look better. Just show predicted vs observed and quantify the difference! | The thresholds are deliberate arranged to assess the flood impacts. Please see the responses to specific comments #14 for details. |
| Figure 3 | | Their workflow doesn't make a lot of sense and doesn't follow the same sequence/layout as most other people who use Pix4D or Agisoft Metashape for SFM. Also, how do they claim to use only 3 GCPs for 'Point cloud with absolute 3D coordinates', then at the next step also do 'Accuracy assessment'? Independent GCPs from those used for georeferencing are needed for accuracy assessment. | The Pix4D workflow of data processing is identical to the suggestions in references. We have added more discussion on the GCP requirement in the revised manuscript. |
| Figure 4 | | This is irrelevant to the study. They have not independently implemented these algorithms, but are just using Pix4D, so no point showing any diagrams like this. | Figure 4 has been deleted. |
| Figure 5 | | Motorway takes up a large part of the DEM, as does vegetation. It is not at all clear how this is accounted for after it is 'removed'. The 3 GCPs are not enough, nor are they properly distributed throughout the study domain. There are unknown edge effects in the orthomosaic/DEM. Usually a UAV is set to fly a regular grid with zig-zag lines with 80% front overlap of images and 60% | The freeway is elevated and supported by the pillars at the centerline of the Keelung Road. Since the elevations of the pillars are higher than the surrounding road surface, it has no impact on the flow. The freeway is removed for CFS because flood water is allowed to flow across underneath the viaduct. Since the ground elevations were observed by the UAV from the divisions between the two viaduct lanes and those between viaducts and the buildings on the roadsides, the DEM underneath the conduct can be estimated. As for the overlap rate, the front |

| | | side overlap of images (more overlap is better). This then generates a DEM and orthomosaic where the edges are low accuracy (due to insufficient overlap), with edge areas being cropped out of the final orthomosaic and DEM. Here there is a strange scattering of points and rough boundaries at the edges of the orthomosaic which raises questions about the accuracy of the orthomosaic, DEM and the UAV flight paths used. The orthomosaic and DEM are cropped in figure 6 and beyond (which is good), however the anomalies in figure 5 are not accounted for. | overlap is 85% while side overlap is 75%. The coverage of each photo is shown in the following figure. We have added these information in the revised manuscript.  Fig. 7 The coverage of UAV photos |
|---|---|---|---|
| Figure 6 | | Check EXIF information for photo capture time. This information may be scrubbed from images automatically by PTT, but is worth checking. Photo locations should be surveyed with RTK GPS and depth measured with a ruler by comparing water level on reference objects such as walls, buildings, bike tires etc. | The photos acquired from PTT were not the original photos and the EXIF information were not available. |
| Figure 7 | | The colour scheme and graduation does not resolve the finer scale features needed for CFS. It would be better with a logarithmic scale. Or just from 5-6 m and buildings will all be one colour. Potentially the colour scheme was selected to hide a poor quality underlying DEM. | The color scheme has been changed to highlight the details in DEM with different resolutions. |
| Figure 8 | | Again poor selection of DEM scale. Lumped flood bins used | The scaling of the figure has been adjusted to highlight the DEM details on ground level. The |

| | | rather than a continuous colour bar. Why? To hide problems? Or just poor choice of data representation? Where are the sewers and manholes? How are they accounted for? Why did they choose to run the study in an area where the motorway blocks so much of the computational domain? | scaling of flood depth is arranged on specific purposes. The figures displayed sewer and manholes are added. Details can be found in the responses to specific comments #7, #13, and #14. |
|---|---|---|---|
| Figure 9 | | Validation? Upstream flow into computational domain? Which is better? Results of 0.5m or 5m simulation? No real way to prove it as no external validation. The VGI data is hardly proof. Even if 0.5m grid is more accurate (as everyone expects) this is not news. Finer grid usually gives better computational results. | Figure 9 shows the comparison of CFS results with different DEM resolutions. The validation of CFS results is not the point here because the flood model has been validated elsewhere in previous papers. In fact, the CFS results in both cases show good agreement with the VGI photos. Indeed, finer grid gives better CFS results is a common sense, but how to prove it is another story. The founding in this study is symbolic because it is the first time CFS can actually be conducted with 0.5m DEM resolutions with the aid of UAV and demonstrate its strength in considering building sidewalls and terrain depressions on water transport. |

**Further specific suggestions:**

| Section | Suggestion | Response |
|---|---|---|
| Abstract | Quantify the accuracy, rather than saying 'more accurately'. | Revised accordingly. |
| 1 Introduction | DEM resolution is important, but the proper representation of sub grid scale features is often more important (e.g. wall, stop-banks, culverts, bridges etc). How these are represented in a coarse DEM is critical. Multi resolution DEMs are possible. Also discuss how roughness is parameterised. I.e. if a modelling cell contains vegetation vs rocks vs concrete. This is also relevant at the end of the results | The considerations of wall and bridge pillars and the roughness parameterization can be found in the responses to specific comments #3, #4, and #8. The computational time for the CFS are 1,127 mins and 16 mins (with Intel I7 processor at 4.2 GHz) for the cases with 0.5m and 5.0m grid size, respectively. |

| | section where it talks about computational efficiency and grid resolution. | |
|---|---|---|
| 1 Introduction | Discusses DEMs from UAVs and LIDAR. See technical correction above about explicitly stating 'UAV aerial imagery'. LIDAR can also be flown on UAVs. | Revised accordingly. |
| 4 Conclusions | 4 Summary and conclusions | Revised accordingly. |

---

## Author Comment (AC2) · 26 May 2020

**Responses to Anonymous Referee #2**

**MS No.: hess-2020-59**

**General comments**

The authors present a case study where UAV and VGI photos are used to run a CFS including a coupling between surface and sewer system flows and they evaluate the influence of the DEM resolution on simulation results. The UAV images are processed to create a high resolution DEM. Results of the numerical simulation are compared to VGI photos in terms of water depth to assess its correctness. A comparison is made between 2 DEM resolutions, and the conclusion is that higher resolution yields better results. The ideas developed in this paper are interesting, and are a useful contribution towards simple and flexible hydraulic numerical simulation by using remote sensing and crowdsourcing. Nevertheless, the paper has to be improved in many points. There is a global lack of scientific rigor, of precision. Many important details are not mentioned, whereas some parts are out of the scope of the paper. Furthermore, the overall English quality is poor. Because of all the weaknesses, both on form and content, I suggest this paper should not be accepted for publication.

**Response:** The paper has been thoroughly revised by adding necessary details and correcting the errors in English according to the reviewer's comments point by point. Hopefully, these revisions will make this paper more acceptable for publication.

**Specific comments**

Many remarks are written in the technical corrections part, but in this part I sum up comments on three major issues:

2.1 DEM generation

This section should be totally rewritten. General considerations about DEM generation and application to the case study are mixed, and not put in a correct and logical order. Camera calibration should be addressed before absolute positioning of images. The authors should consider being shorter on generalities and give more information about their own input during this step. The authors do not say how they replace the DEM parts they remove: motorway bridge and vegetation. How is the new altitude chosen? Don't bridge pillars or vegetation have an influence on the surface flow? How do the authors apply the method from Rabatel et al. without removing the NIR filter of the camera in the first place?

**Response:** Thanks for the reviewer's suggestion. The Section 2.1 has been reorganized by moving forward the camera calibration before image positioning and adding the instructions on how to generate the DEM and consider the roughness after removing the viaduct and vegetation regions (please see the responses to Technical corrections #12 and #13 for detail). Since the elevations of the pillars are higher than the surrounding road surface, it has no impact on the flow. The removal of NIR filter is not necessary because an ExG-ExR binary index is applied to detect the region of vegetation. To prevent confusion, the citation of Rabatel et al.'s method has been removed (please see the response to Technical corrections #15 for detail).

**2.2 CFS model**

Very little information about the model is given in the paper.    The reader is referred to 2 other papers (Jang et al. 2018 & 2019).    The effort made not to be redundant is appreciated, but some more details are needed about: 1) the numerical coupling 2) the manholes location and implementation in the model 3) the chosen hydraulic parameters, roughness distribution, boundary conditions. . . 4) the time extent of the simulation.

**Responses:**

(1) When rain drops, the OFM is firstly initiated for surface water routing. Then, the surface runoff travels for a distance and enters the sewer pipes via the street inlets to trigger the SFM. When the sewer pipes get full, the sewer water surcharges back onto ground surface via the manholes or inlets. In the simulation process, the water exchanges between the two models are realized by adding a source/sink term in the continuity equations which can be determined by weir and orifice functions via one-to-one relationship as shown in the schematic diagram Fig. 1.

[Figure]

Fig. 1 Illustration of the water simulation process between ground surface, inlet, sewer pipe and manhole.

(2) The pipelines and manholes of the sewer systems are obtained from governmental filed survey data and are displayed in Fig. 2.

[Figure]

Fig. 2 The pipeline and manholes of the sewer systems in the study area.

(3) The weir coefficient and the orifice coefficient are set as 0.48 and 0.57, respectively, according the suggestion by Lee et al. (2013). The Manning's coefficient n is used to represent the surface roughness subject to land covers. Since the land covers are mostly concrete and short grass, the value of n slightly varies from 0.03 to 0.04 according to Chow, V.T. (1959). Although the skin friction represented by Manning's n changes little, the form frictions caused by road curbs and building walls can be more significantly presented as DEM resolution increases. At the boundary of the simulation domain, the water is allowed to outflow freely.

(4) The time extents for the CFS are 1,127 mins and 16 mins (with Intel I7 processor at 4.2 GHz) for the cases with 0.5m and 5.0m grid size, respectively.

**Reference:**

Lee, S.S., Nakagawa, H., Kawaike, K., and Zhang, H., 2013. Experimental validation of interaction model at storm drain for development of integrated urban inundation model, J. Jap. Soc. Civil Eng. Ser. B1 (Hydraulic Engineering), 69(4), I_109-I_114.

Chow, V.T., Open-channel hydraulics. McGraw-Hill, New York, 1959.

**2.3 Results**

The DEM accuracy is checked on the same points that were used to do its absolute georeferencing, which does not prove anything. The authors' assertion that coarser resolution leads to lower quality results is not confirmed by the presented arguments. Water depths of both simulations are compatible with all the VGI photos. Water depths on rooftops are not measured nor observed (or some information is missing in the paper). Flood duration differs between resolutions, but none is contradicted by some observations. The conclusion concerning the DEM resolution is not supported enough to be stated as it is.

**Response:** For the issue of DEM accuracy, please refer to the specific comments #24. Our study indicates that the building sidewalls and terrain depressions have a great influence on flood extent, depth, and occurrence which can only be simulated by the CFS with high-resolution DEM. Please see the response to the specific comments #35 for detail. The flood duration issue, please refer to the specific comment #32. We also revised the conclusion concerning the DEM in the revised manuscript.

**Technical corrections**

1.  title . . . to sophisticated urban
    l.15 . . . flash flood event that occurred on
    l.17 . . . network are used to establish
    l.18 . . . data is resampled
    l.21 . . . and VGI lowers the
    l.26 Flash floods resulting from extreme rainfall

    **Response:** Revisions have been made according to the suggestions above.

2.  l.28-29 & Table 1: What is the point of this table? Is it really needed in this paper?

    Line 29 mentions life losses, but no figures about it in the table. The authors should consider deleting this table and modifying the text accordingly

    **Response:** Table 1 and relevant texts have been removed from the revised manuscript.

3. l.52-53 Citations only refer to VGI, not UAV techniques

**Response:** Thanks for pointing this out. The citations have been moved to the paragraph where only VGI is mentioned as the following:

"The VGI considers every citizen as a sensor to acquire spatial data on a wide range of phenomena via crowdsourcing the keywords on social media such as Facebook, Twitter, Instagram, etc. (Le Coz, et al., 2016; Michelsen, et al., 2016; Starkey et al., 2017; Tauro et al., 2018; Goodchild and Glennonm, 2010)."

4. l.63 . . . can be used as

**Response:** Revised accordingly.

5. l.63-64 Does "boundary conditions" mean "hydraulic boundary conditions"? More explanations are needed here to understand the link between boundary conditions and spatial resolution

**Response:** The original statements is not clear and has been revised as "The DEM generated from UAV provides detail terrain of an urban area which significantly increases the spatial resolution of CFS compared to traditional practices."

6. l.69 . . . event that occurred on
l.70 . . . occurred between 13:00 and 18:00
l.71 . . . 131.5 mm/h from 14:30 to 15:30
l.72 . . . rainfall intensity exceeded
l.73 . . . Chanxing street near the National
l.75 VGI are used to establish

**Response:** Revisions have been made according to the suggestions above.

7. l.79 "finally, the simulated results are compared" Contradiction with fig. 3 where VGI seems to be an input to the CFS.

**Response:** Thanks for pointing this out. The revised figure is shown in Fig. 3.

[Figure]

Fig. 3 Conceptual flowchart of this study (the VGI photo was adopted from PTT, Taiwan)

8. Section 2.1 This section should be totally rewritten. DEM generated General considerations about DEM generation and application to by UAV the case study are mixed and not put in a correct and logical order. Camera calibration should be addressed before absolute positioning of images. The authors should consider being shorter on generalities and give more information about their own input during this step

   **Response:** Thanks reviewer's suggestions. The authors have reorganized Section 2.1 by adding the instructions on how to generate the DEM and consider the roughness after removing the viaduct and vegetation regions. The part of camera calibration has been moved forward to the part of image positioning. Meanwhile, the flight regulations for conducing the UAV survey are also mentioned in the revised Section 2.1.

9. l.82 . . . left side of Fig. 3
   l.82-83    . . . methods for generating a DEM
   l.84 . . . 2004). They are based
   l.86, 92, 93    . . . coordinates
   l.90  The six parameters [. . .] are determined during exterior orientation

   **Response:** The manuscript has been revised according the above suggestions.

10. l.96  What resolution is chosen for resampling?

    **Response:** After space intersection, the average ground sampling distance of point cloud is 0.03m. The UAV images were processed to generate orthomosaic image and digital surface model (DSM) with Pix4Dmapper Pro Version 1.4.46 (Pix4D). The orthoimage and the DEM are then resampled under spatial resolutions of 0.5 m and 5 m for CFS.

11. l.100      . . . condition from 06:00

    **Response:** Revised accordingly.

12. l.117  Vegetation is removed: how the new ground altitude is computed? With which roughness? Does vegetation have no impact on the flow? How the results are biased by this removal?

    **Response:** The regions of vegetation are detected using the ExG-ExR binary index (Meyer and Neto, 2008) by subtracting the alternate excess red vegetative index (ExR = 1.4r−b) from the excess green vegetation index (ExG = 2g−r−b), where r, g, and b are the chromatic coordinates. To consider the friction effects by the roughness of vegetation, the Manning's coefficient is set as 0.04 for CFS.

    **Reference:** Meyer, G. E. and Neto, J. C.: Verification of color vegetation indices for automated crop imaging applications. Computers and electronics in agriculture, 63(2), 282-293. https://doi.org/10.1016/j.compag.2008.03.009, 2008.

13. l.117The viaduct is removed: how the new ground altitude is computed? With which roughness? Do the viaduct's pillars have no impact on the flow?

    **Response:** The freeway is elevated and supported by the pillars at the centerline of the Keelung Road. Since the elevations of the pillars are higher than the surrounding road surface, it has no impact on the flow. The freeway is removed for CFS because flood water is allowed to flow across underneath the viaduct. Since the ground elevations were observed by the UAV from the divisions between the two viaduct lanes and those between viaducts and the buildings on the roadsides, the DEM underneath the conduct can be estimated. For CFS, the roughness are estimated by setting

Manning's coefficient equivalent to 0.035 according to Chow, V.T. (1959). The discussions above have been added to the revised manuscript.

**Reference:** Chow, V.T. (1959) Open Channel Hydraulics. McGraw-Hill, New York.

14. l.118. . . shrubs and grass

    **Response:** Revised accordingly.

15. l.120-121 The authors did apparently not fully understand the publication by Rabatel et al., which needed a removal of the NIR blocking filter inside the camera. This paper does not mention this removal, nor gives any detail about the specific linear combination.

    **Response:** Thanks for pointing this out. Since the regions of vegetation are detected using the ExG-ExR binary index (Meyer and Neto, 2008) by subtracting the alternate excess red vegetative index (ExR = 1.4r−b) from the excess green vegetation index (ExG = 2g−r−b), where r, g, and b are the chromatic coordinates, and there is no need for removing NIR blocking filter inside the camera. To prevent confusion, the citation of Rabatel et al.'s method has been removed.

16. l.121-122 Some of the surrounding buildings must have heights above the 9 m threshold. However the authors claim that they remove the viaduct, but not the buildings (l. 124-125)

    **Response:** The distances between the viaduct and the surrounding buildings are about 10m, so the elevation information of the buildings can be retained. This missing explanation has been added.

17. l.124        . . . flow smoothly on ground surface

    **Response:** Revised accordingly.

18. l.125        Transverse is not the adequate word

    **Response:** The word "transverse" is revised as "traverse".

19. l.125        There is no mention of (parked) cars and street furniture. They might have an influence on the flow. Are they in the DEM or not? The authors should give a comment on this.

    **Response:** UAV is applied to collect images at off-peak traffic time (6-7 am), and UAV images show no parked cars and street furniture in the experimental area (Keelung Road). Terrain models built by UAV only reflect the static and stable ground features, which is why UAV-based DSM and orthophotos only show roads (sign lines), buildings and vegetation.

20. l.133        . . . these photos are used to

    **Response:** Revised accordingly.

21. l.135        The described hydrological event lasts for a few dozens of minutes. A slight shift in the image timestamp could lead to totally wrong information. The authors should only take into account photos for which the timestamp corresponds to the moment when it was taken, not when it was posted on a social network.

    **Response:** The VGI photos acquired from internet, for example from the PTT in this study, are usually not the original photos and therefore the EXIF information is not available (We have checked it using http://metapicz.com). However, the differences between the timestamp of VGI

photos and the time they were taken should not be significant due to the timeliness requirement on social media.

22. l.135-136 The authors should give an idea of uncertainties yielded by manual flow depth estimation. What about an automatic water level estimation?

    **Response**: The yellow triangles in Fig. 6 show the reference positions for flooding. Based on the elevation of the ground, road curbs, and buildings and wheel size of bikes, the observed flooding depths are estimated. The depth of 0.05m (about the ankle height) is used as a reference indicator to show the flood situation, which in a way includes the uncertainty of the water depth estimation.

23. Section 2.3 This section lacks details about the chosen hydraulic parameters CFS model (roughness, boundary conditions) for both OFM and SFM. It is not clear how the manhole positions are determined: DEM and water levels are obtained from images, why not the manholes? It is also not clear which type of interaction between OFM and SFM is applied. The authors should also give the time limits of their simulations.

    **Response**: The pipelines and manholes were obtained from the field survey data provided by the local government. Please see the responses to the specific comments about Section 2.2 for details.

24. l. 153 The DEM accuracy is checked on 3 GCPs. The same 3 GCPs were used to perform the DEM georeferencing (see lines 104-108). The authors should use a different set of control points than those who were used to process the data, with a better spatial distribution in the modeled area. Furthermore, 2 DEM are created, with distinct resolutions. Accuracy should be checked for both.

    **Response**: The coordinates of the three GCPs were obtained by referring to the publicly released values of Taipei City Government and using the static positioning of Global Navigation Satellite System (GNSS) with positional accuracy in centimeter level. The difference between the coordinates obtained by these two methods can be used to evaluate the accuracy of the ground control points. The 0.5m and 5m DEMs are created and validated according to the initial UAV-based DEM with resolution of 0.03m. We have added more discussion on the GCP requirement in the revised manuscript. The reasons we used only three GCPs are (1) the study area is relative small ($0.0637 \text{ km}^2$) and the GPS information on the UAV could produce 3D coordinate with certain degree of accuracy; (2) there are exactly three GCPs released by the Taipei City Government in this study area and we also double check the released values with the static positioning of Global Navigation Satellite System (GNSS) with positional accuracy in centimeter level; (3) The number of GCP depends on the surveying areas, flight altitudes, resolutions and application goals. According to the user manual of Pix4D (https://support.pix4d.com/hc/en-us/articles/204272989-Offline-Getting-Started-and-Manual-pdf), a minimum number of 3 GCPs is required.

25. l. 156 The authors did not give the initial DEM resolution

    **Response**: The original DEM resolution is the same as the average ground sampling distance of point cloud equivalent to 0.03m.

26. l. 159     . . . simulation, the grid meshes

    **Response**: Revised accordingly.

27. l. 167-169     The authors state that water on rooftops is better simulated by the fine resolution model. Could they give some validation criteria to explain this statement? Could they explain why water should be on rooftops? Does their model take water evacuation from rooftops into

account?

**Response**: The water accumulated on rooftop because there are usually parapet walls with about 1 meter height on the rooftops around the borders of buildings in Taiwan. When the DEM resolution is high enough, the elevations of parapet walls can be represented by the grid-based mesh system in CFS and the water detention on the rooftops can be simulated. The rougher the grid/DEM resolutions, the faster the stored water will evacuate through the gaps between two adjacent grid cells. This discussion has been added to the revised manuscript.

28. l.173      Higher: not much higher. The authors should give the values.

**Response**: Revised accordingly.

29. l.174      Lower resolution implies that small terrain features are not represented, so the topography should be smoother. The authors observe the opposite. The authors should check the DEM resampling, and give more comments about their observation.

**Response**: Because the ground levels are given in grid unit in CFS, there exists an invisible wall between two adjacent grids with different elevations. When DEM resolutions decrease, these walls become higher which result in larger blocking effects that reduce inter-cell water communications. This phenomenon explains why the flood water travels slower in the simulated results. Relevant discussions have been added.

30. l.178      . . . results. It can be seen

**Response**: Revised accordingly.

31. l.180      . . . At the remaining points, the simulated

**Response**: Revised accordingly.

32. l.184-186 I cannot see the causal relationship between the fact that there are no VGI photos after 15:40 and the overestimation of flood duration. Since there are no photos, it is not possible to conclude in a way or another.

**Response**: Indeed, to be more precise, the original statement has been modified as "At point #2, the flood duration may be overestimated under 5 m DEM resolution because the flood depth at point #2 does not decrease with time at all even when the rainfall has stopped for one hour."

33. l.195      . . . than that that. . ..

**Response**: Revised accordingly.

34. l.207-208 DEM updates, and resulting CFS are not in real time, since weather conditions generally prevent UAVs to fly during or just after heavy rainfalls.

**Response**: The original sentence has been modified as "Aided by the rapid growing technologies of remote sensing and crowdsourcing, it is possible to efficiently update DEM data and record the flood depth by UAV and VGI in a short time after heavy rainfalls".

35. l.211 It has not been demonstrated in the paper that fine resolution modeling results are better. The comparison presented in table 4 shows that both resolutions give results consistent with VGI photos. Other validation data is needed to be able to draw this conclusion on this case study.

**Response**: Thank you for the comment. The main attention of this study is to extract useful information by image processing technologies from VGI photos and UAV data for an urban

flooding event. In such events, it is very common that onsite water level observations are unavailable which raises the difficulty of CFS validation. However, from the comparison of CFS results with VGI photos, this study indicates that the building sidewalls and terrain depressions have a great influence on flood extent, depth, and occurrence which can only be simulated by the CFS with high-resolution DEM. The original statements have been revised according to the above discussions.

36. Table 1 What is the point? The list of flash floods that occurred in 2019 is not relevant to the paper. Moreover, events after July, 8th are not listed.

**Response**: Table 1 has been deleted and the relevant sentences in the Introduction have been rewritten.

37. Table 3 The accuracy is checked on the very same points used in the DEM production process. Low error values are thus expected. It gives no information on the accuracy of the DEM everywhere else.

**Response**: The coordinates of the three GCPs were obtained by referring to the publicly released values of Taipei City Government and using the static positioning of Global Navigation Satellite System (GNSS) with positional accuracy in centimeter level. Since the two coordinates are determined from different methods, they can be compared to evaluate the accuracy of the ground control points.

38. Figure 1 Caption: rain gauge (as in the text). In the bar chart, times are written above bars, I guess they should lie between bars

**Response**: Thank you for the comment. The term "rain gauge" and the revised figure is shown in Fig. 4.

[Figure]

Fig. 4 The rainfall hyetograph on 14 June 2015 at GongGuan rain gauge station (C1A760).

39. Figure 2 Rain gauge (as in the text). Top left image is not necessary. I do not understand the gray levels in top right image : shade or altitude? Bottom image is very dark, authors should improve brightness and contrast. The cartographic scale is indicated only for the bottom image Figure 3. The right part is not consistent with the paper. VGI seems to be an input of CFS. Parameter of Drainage system: is there only one parameter? This parameter is not even mentioned in the paper.

**Response**: Thank you for the comment. Rain gauge has been corrected.

The Top left image is aiming to show the location of Taipei City in Taiwan. The gray levels in top right image shows the hillshade as it presented in the legend of this image. The bottom image has been adjusted to improve its brightness and contrast. A cartographic scale is also added to all the images (please see Fig. 5). The VGI is not an input of CFS so that the Fig. 3 is revised (see the response to Technical corrections #7). The parameters of the drainage system have been added (Please see the responses to the Specific comments about Section 2.2).

[Figure]

Fig. 5 Study area (red polygon) at GongGuan, Taipei, Taiwan (the Google Earth images sourced from © Google, Landsat/Copernicus and the DEM in the top left image and the hillshade in the top right image were derived from SRTM with 30 meter resolution).

40. Figure 4   Not necessary in this paper

    **Response**: The original figure has been removed.

41. Figure 5   The caption reads: "images taken", but they must have undergone some processing. The authors should tell readers what is exactly displayed in fig. 5. The image is distorted compared to the area shown in the following figures (stretched along Y axis). Why? GCP #1 is located just next to the viaduct, so there is a discontinuity in altitude just nearby. It is recommended to select clear zones for GCPs

    **Response**: Thank you for the comment. The caption has been revised. We noticed that the image is distorted and it has been corrected (see Fig. 6). Since GCP #1 is about 10 m away from the viaduct, there is enough room to display the continuity of altitude.

[Figure]

Fig. 6 Images taken by UAV and the distribution of the ground control points

42. Figure 6   All minutes in timestamps seem to be rounded to the nearest 10. Why? How water depths could possibly be estimated from photo #1?

    **Response**: Thank you for the comment. Because all the photos were collected from internet and meanwhile the time step for rainfall observation and flood simulation is 10 minutes. So we added the timestamps for all photos to the closest 10 minutes before the photos were posted online. In photo #1, we noticed that the water level is very close to the surface of sidewalk along the road and the sidewalk is generally 0.1 m higher than the road in Taiwan.

43. Figure 7   The choice of the color scale is not adequate to allow a good perception of low value altitudes (everything is blue). Moreover, the rainbow color scale should be avoided: https://www.nature.com/articles/519291d  https://www.climate-lab-book.ac.uk/2014/end-of-the-rainbow/

    **Response**: We have changed the color scale of this figure and also checked the image on the Coblis (https://www.color-blindness.com/coblis-color-blindness-simulator/). The revised figure

is shown in Fig. 7.

[Figure]

(a)            (b)            (c)

Fig. 7 (a) Orthoimage; (b) the DEM with spatial resolution of 0.5 m; (c) the DEM with spatial resolution of 5 m

44. Figure 8  Water depths between 0.00 and 0.05 m should appear in white according to the caption. I can not see any white pixel. Why? The difference between the colors corresponding to 0.15-0.30 and 0.30-0.50 water depths is too small to be distinguished. Water accumulates on roofs, especially for the fine resolution model. It seems very unlikely, especially for the building lying in the left.

**Response**: Thank you for the comment. The depths between 0.00 and 0.05 m should appear as no color. We have corrected it and adjust the scaling of the figure to highlight the DEM characteristics on road surface (the revised figure is shown in Fig. 8). The difference between the colors corresponding to 0.15-0.30 and 0.30-0.50 water depths has been adjusted.

[Figure]

[Figure]

[Figure]

Fig. 8 Simulated flood extents at different time under DEM resolution 0.5 m (left) and DEM resolution 5 m (right).

The water accumulated on rooftop is because there are solid parapet wall with about 1 meter height around the borders of these buildings. For example, it just like the parapet wall in Fig. 9.

[Figure]

Fig. 9 Parapet wall around the border of building. (Image credit: good.ruten.com.tw)